# scMultiome analysis identifies embryonic hindbrain progenitors with mixed rhombomere identities

Yong-Il Kim, Rebecca O'Rourke, Charles G Sagerström*

Section of Developmental Biology, Department of Pediatrics, University of Colorado Medical School, Aurora, United States

**Abstract** Rhombomeres serve to position neural progenitors in the embryonic hindbrain, thereby ensuring appropriate neural circuit formation, but the molecular identities of individual rhombomeres and the mechanism whereby they form has not been fully established. Here, we apply scMultiome analysis in zebrafish to molecularly resolve all rhombomeres for the first time. We find that rhombomeres become molecularly distinct between 10hpf (end of gastrulation) and 13hpf (early segmentation). While the embryonic hindbrain transiently contains alternating odd- versus even-type rhombomeres, our scMultiome analyses do not detect extensive odd versus even molecular characteristics in the early hindbrain. Instead, we find that each rhombomere displays a unique gene expression and chromatin profile. Prior to the appearance of distinct rhombomeres, we detect three hindbrain progenitor clusters (PHPDs) that correlate with the earliest visually observed segments in the hindbrain primordium that represent prospective rhombomere r2/r3 (possibly including r1), r4, and r5/r6, respectively. We further find that the PHPDs form in response to Fgf and RA morphogens and that individual PHPD cells co-express markers of multiple mature rhombomeres. We propose that the PHPDs contain mixed-identity progenitors and that their subdivision into individual rhombomeres requires the resolution of mixed transcription and chromatin states.

## Editor's evaluation

This study transcriptomically profiles the developing zebrafish hindbrain from gastrulation through stages of rhombomere formation in the zebrafish embryo. The transcriptomic data is very thorough and will be a valuable resource to the field.

*For correspondence: charles.sagerstrom@cuanschutz. edu

Competing interest: The authors declare that no competing interests exist.

## Introduction

Formation of a functioning central nervous system requires that neural progenitors are properly specified and positioned during embryogenesis. One model posits that neural progenitors emerge in segmental compartments (termed neuromeres) that are formed by subdivision of the embryonic neural primordium along its anteroposterior (AP) axis (*Keynes and Lumsden, 1990*). While it remains under debate whether this model applies to the embryonic fore and midbrain, the embryonic hindbrain is clearly divided into neuromeric units known as rhombomeres (*Lumsden, 1990*), but the unique characteristics of each rhombomere have not been defined molecularly and it remains unclear how they form during embryogenesis.

It has long been noted that the hindbrain possesses a two-rhombomere periodicity where each rhombomere pair provides neural crest cells and motor innervation to one specific pharyngeal arch (*Lumsden and Keynes, 1989*; *Lumsden et al., 1991*). Each pair consists of one odd- and one even-numbered rhombomere that display unique properties – e.g., neuronal differentiation generally occurs

earlier in the even-numbered rhombomere. Transplantation experiments further revealed that cells from odd versus even rhombomeres do not intermix but form discrete boundaries when they come into contact (*Guthrie and Lumsden, 1991*). Strikingly, cells from the even-numbered member of one pair intermix more readily into other even-numbered rhombomeres than into the odd-numbered member of the same pair (*Guthrie et al., 1993*). Individual rhombomeres are, therefore, characterized by belonging to a specific rhombomere pair as well as by possessing either an odd or an even identity. This periodicity is recapitulated in some gene expression patterns. For instance, *egfl6* is expressed in even-numbered rhombomeres (*Thisse et al., 2001*; *Choe et al., 2011*; *Ghosh et al., 2018*) and *epha4* in odd-numbered ones (*Xu et al., 1995*), while *mafba* is expressed in a two-rhombomere unit (r5/r6). r5 and r6 can, therefore, be assigned to the same rhombomere pair by their shared expression of *mafba* and to an even (*egfl6* expression in r6) versus odd (*epha4* expression in r5) identity, but similar assignments have not been possible for other rhombomeres. Indeed, relatively few genes have been identified that recapitulate this periodicity. It is, therefore, unclear how rhombomeres differ in terms of gene expression programs and whether an odd versus even periodicity can be defined molecularly.

It also remains unclear how the rhombomeres emerge from the hindbrain primordium. Visual observations in embryos of various species suggest that rhombomeres develop from broad early segments of the neural tube that are subsequently subdivided. For instance, 'pro-rhombomeres' have been observed in the chick (*Vaage, 1969*) and 'primary' rhombomeres have been proposed in the human (*Müller and O'Rahilly, 1997*; *Müller and O'Rahilly, 2003*). In the zebrafish, the r3/4 and r4/r5 boundaries form first – thereby delineating r4 – followed by the r6/7 and r1/r2 boundaries that demarcate the r2/r3 and r5/r6 segments. Limited genetic data support the existence of a two-segment pattern in the early hindbrain. Zebrafish *valentino* (*val*) embryos carry a mutation in the *mafba* gene that is expressed in r5 and r6 (*Moens et al., 1998*; *Moens et al., 1996*). In *val* animals, r5 and r6 do not form, but are replaced by an 'rX' segment that displays mixed r5/r6 identity. This finding was interpreted to mean that rX represents an early 'protosegment' containing progenitors for r5 and r6, but it is unclear if rX cells exist in wild-type embryos and there is no genetic or molecular evidence suggesting the presence of equivalent progenitor domains for other rhombomeres.

Attempts at better defining rhombomere identities and understanding their formation have been hampered by a lack of comprehensive molecular data, as recent scRNAseq analyses could not fully resolve individual rhombomeres (*Jiang et al., 2021*; *Farnsworth et al., 2020*; *Wagner et al., 2018*; *Tambalo et al., 2020*; *Farrell et al., 2018*). We reasoned that scMultiome analysis, which combines RNAseq and ATACseq of individual nuclei, might provide an improved definition of rhombomeres. Applying this method to zebrafish, we were able to resolve all rhombomeres for the first time. We find that rhombomeres become molecularly distinct between 10hpf (end of gastrulation) and 13hpf (early segmentation) and remain distinct at 16hpf (mid-segmentation). We do not detect extensive similarities among odd versus even rhombomeres, revealing an absence of general molecular identities defining these two states at early embryonic stages. Instead, we find that each rhombomere displays a unique gene expression and chromatin profile. Prior to the appearance of distinct rhombomeres, our scMultiome analysis identified three cell clusters (HB.1–3) that we refer to as primary hindbrain progenitor domains (PHPDs). Based on our analyses, these PHPDs represent r2/r3 (possibly including r1), r4 and r5/r6. We note that these domains correlate with the earliest visually observed segments in zebrafish embryos and propose that the HB.2 PHPD corresponds to the rX segment identified in *val* mutants. We further find that PHPD cells exhibit mixed gene expression profiles, such that individual cells co-express markers of multiple rhombomeres, and we verified the existence of such mixed-identity cells in vivo. Lastly, we demonstrate that the retinoic acid (RA) and fibroblast growth factor (Fgf) morphogens are required for the formation of the PHPDs. We propose that rhombomeres arise from PHPDs in a process where mixed transcriptional identities are resolved into specific rhombomere identities.

## Results
### Individual rhombomeres are molecularly resolved at segmentation stages

We applied scMultiome analysis to dissected hindbrain regions at 16hpf (when rhombomeres are becoming morphologically observable; *Hanneman et al., 1988*) and at 13hpf (when

rhombomere-restricted gene expression is well-established), as well as to whole embryos at 10hpf (the end of gastrulation). After data processing (see Methods and *Figure 1—source data 1*), unsupervised clustering followed by projection as UMAPs revealed multiple distinct clusters at each timepoint (*Figure 1—figure supplement 1*, *Figure 2—figure supplement 1*, *Figure 3—figure supplement 1*, *Figure 7—figure supplement 1*). We generated lists of differentially expressed genes for each cluster (*Figure 1—source data 2*) and, employing a combination of GO term analysis and comparisons to known tissue-specific genes from the literature and the zebrafish database (*Bradford et al., 2022*), we assigned cell type identities to all clusters. Since our samples contain tissues adjacent to the hindbrain primordium, we find clusters corresponding to multiple cell types, but the most prominent group of clusters at each timepoint contained CNS cells. Based on this initial analysis, we bioinformatically isolated CNS-related cell types (excluding placodal cells and post-migratory neural crest cells but including pre-migratory neural crest cells that could not be readily separated), re-clustered them, and projected them as UMAPs. These neural UMAPs reveal well dispersed clusters at 13hpf and 16hpf, whereas clusters are more tightly grouped at 10hpf (*Figure 1A, B*, *Figure 2A, B*, and *Figure 3A-C*). In agreement with the early stages being analyzed, most cells in these neural UMAPs express *sox3*, a marker of neural progenitors, with clusters of cells expressing genes indicative of neural differentiation (e.g. *neurod4*) emerging at 13hpf and 16hpf (*Figure 1C, D*, *Figure 2C, D*, and *Figure 3L-M*).

At early segmentation stages (13hpf), individual rhombomeres are readily identifiable in the neural UMAP based on their gene expression profiles. The r3, r5, and r6 clusters are immediately recognizable based on the known expression of *egr2b* (restricted to r3 and r5; *Figure 1E*; *Oxtoby and Jowett, 1993*), *mafba* (restricted to r5 and r6; *Figure 1F*; *Moens et al., 1998*), and *sema3ab* (restricted to r5 at this stage; *Figure 1G*; *Roos et al., 1999*). r4 forms a distinct cluster that is characterized by expression of *fgf3* (enriched in r4; *Figure 1H*; *Walshe and Mason, 2003*), as well as *cyp26b1* that is expressed in r4 before gradually expanding to r3 (*Figure 1I*; *Zhao et al., 2005*) – which we confirmed by hybridization chain reaction (HCR) analysis (*Figure 1Q*). A cluster located adjacent to r4 is a strong candidate to contain r2 cells, but there are few known unique markers for r2. Our data reveal that *tgfbr2b* and *vgll3* are highly enriched in this cluster (*Figure 1J and K*). A recent report indicated that *vgll3* is expressed in r2 of *Xenopus* (*Simon et al., 2017*), but this has not been verified in other systems. We, therefore, examined *vgll3* expression in 13hpf zebrafish embryos and find that it is restricted to r2, confirming the identity of this cluster (*Figure 1R*). A cluster found next to r2 is enriched for *irx1b* (*Figure 1L*) and our HCR analysis detected *irx1b* immediately anterior to *vgll3* expression (*Figure 1S*), indicating that this cluster corresponds to r1. We note that one remaining cluster at 13hpf expresses markers indicative of both r1 and r2, but this cluster is also enriched in genes broadly expressed in the dorsal neural tube, such as *zic* family TFs (*Figure 1N*). A closer examination revealed that dorsal (*casz1* and *zic2a*; *Grinblat and Sive, 2001*) and ventral (*ntn1a* and *sp8b*; *Lauderdale et al., 1997*; *Kawakami et al., 2004*) genes are differentially expressed across each rhombomere at 13hpf (*Figure 1M–P*), indicating that dorsoventral (DV) patterning is ongoing at this stage. Since the dissected hindbrain region used for this analysis includes some adjacent neural tissue, the neural UMAP also contains clusters corresponding to the midbrain-hindbrain boundary (MHB), the caudal hindbrain (CHB), the anterior spinal cord (SC), as well as some cells from the midbrain (MB) and forebrain (FB). An analogous analysis of the 16hpf neural UMAP readily identified r1 to r6 (*Figure 2A*). At this stage, we observe two clusters that express r5-specific genes (such as *egr2b*). As at 13hpf, we find that dorsoventral markers are differentially expressed between these clusters and across all rhombomeres (*Figure 2E and F*).

To further confirm the assigned rhombomere identities, we identified ATAC peaks enriched in each cell cluster relative to all other clusters and carried out a de novo search for over-represented transcription factor (TF) binding motifs (*Figure 4*, *Figure 4—source data 1*). At both 13hpf and 16hpf, the r3 and r5 clusters display a strong enrichment for accessible EGR2 motifs (*Figure 4C*; consistent with high expression of *egr2b* in both rhombomeres), but the r3 and r5 clusters also differ from each other such that the r5 cluster is enriched for accessible MAF motifs (*Figure 4D*; in agreement with *mafba* expression in r5) and r3 is enriched in accessible binding motifs for TALE TFs (such as PKNOX1 [a.k.a., Prep1]; *Figure 4E*). The type of TALE binding motif observed in r3 is also enriched in other anterior rhombomeres, and corresponds to a compound binding site that supports binding of TALE heterodimers (Pbx:Prep and/or Pbx:Meis; *Penkov et al., 2013*; *Ladam et al., 2018*), which are broadly expressed in the hindbrain (*Sagerström et al., 1996*; *Zerucha and Prince, 2001*; *Waskiewicz et al., 2001*; *Pöpperl et al., 2000*; *Deflorian et al., 2004*; *Choe et al., 2002*). r6 shows enrichment for

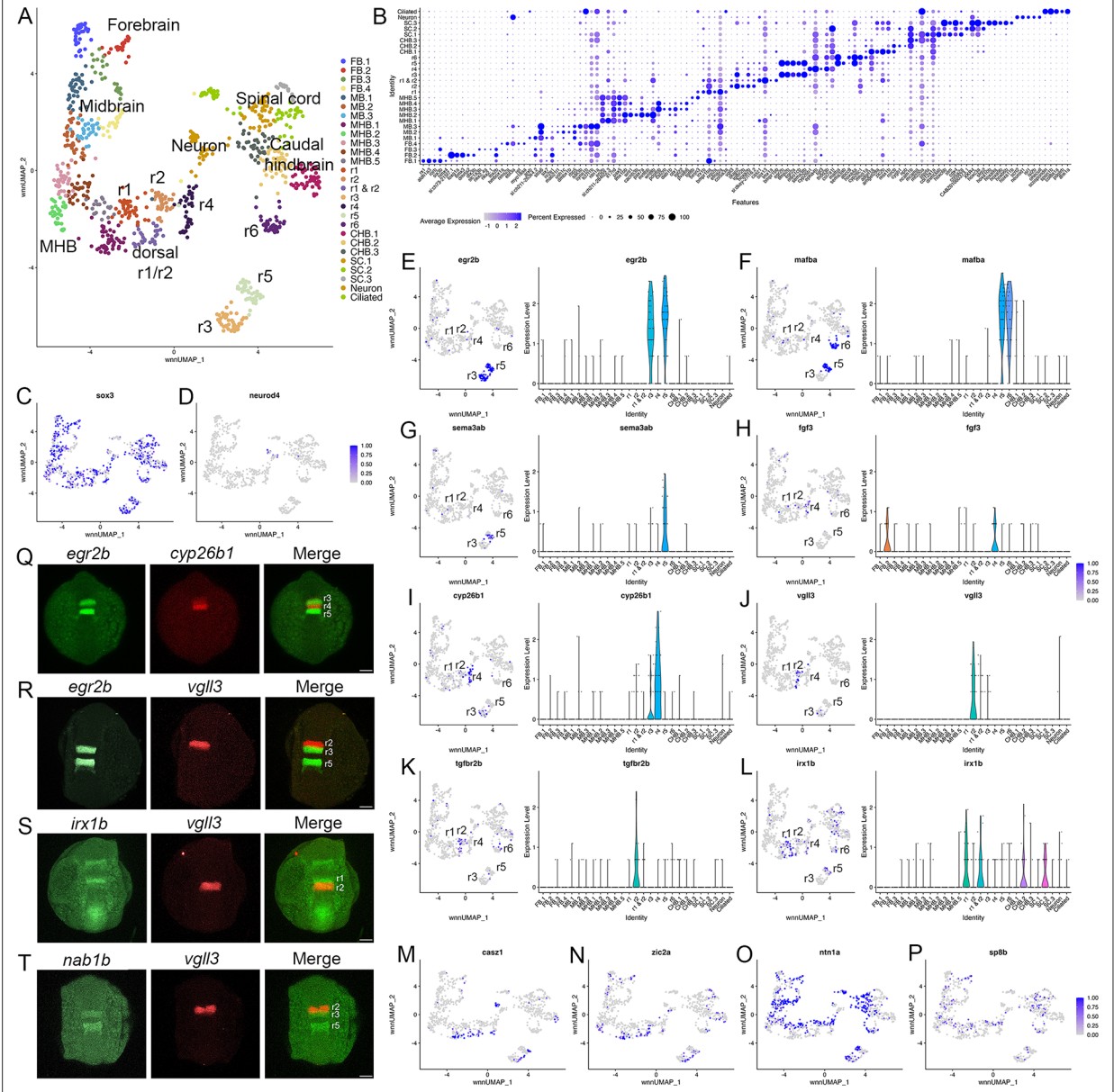

**Figure 1.** Individual rhombomeres are resolved in the 13hpf zebrafish hindbrain. See also *Figure 1—figure supplement 1*, *Figure 1—source data 1* and *Figure 1—source data 2*. (**A**). UMAP of 13hpf neural clusters. (**B**). Dot plot showing the expression of the top five enriched genes in each cluster. (**C**, **D**). Feature plots showing the expressing of *sox3* (**C**) and *neurod4* (**D**). (**E-L**). Expression of rhombomere-specific genes are shown as feature plots (left panels) and violin plots (right panels). (**M-P**). Feature plots showing the expression of dorsoventral marker genes. (**Q-T**). Hybridization chain reaction (HCR) analysis of rhombomere-restricted gene expression in 13hpf wild-type zebrafish embryos. Embryos are shown in dorsal view with anterior to the top. FB = forebrain, MB = midbrain, MHB = midbrain-hindbrain boundary, r=rhombomere, CHB = caudal hindbrain, SC = spinal cord. Scale bar in (**Q-T**) = 100 um.

The online version of this article includes the following source data and figure supplement(s) for figure 1:

**Source data 1.** Quality Control data for scMultiome analyses.

**Source data 2.** Differential gene expression data for each cluster at each time point.

**Figure supplement 1.** scMultiome analysis of 13hpf zebrafish.

accessible MAF motifs (*Figure 4D*; in accordance with *mafba* expression in r6), as well as for paralog group (PG) 4 HOX motifs (*Figure 4F*). PG4 HOX motifs are also enriched in the caudal hindbrain at 13hpf and become less accessible at 16hpf, consistent with the dynamic expression of several PG4 *hox* genes in the caudal CNS with anterior limits at r5 and r6 (*Prince et al., 1998a*; *Prince*

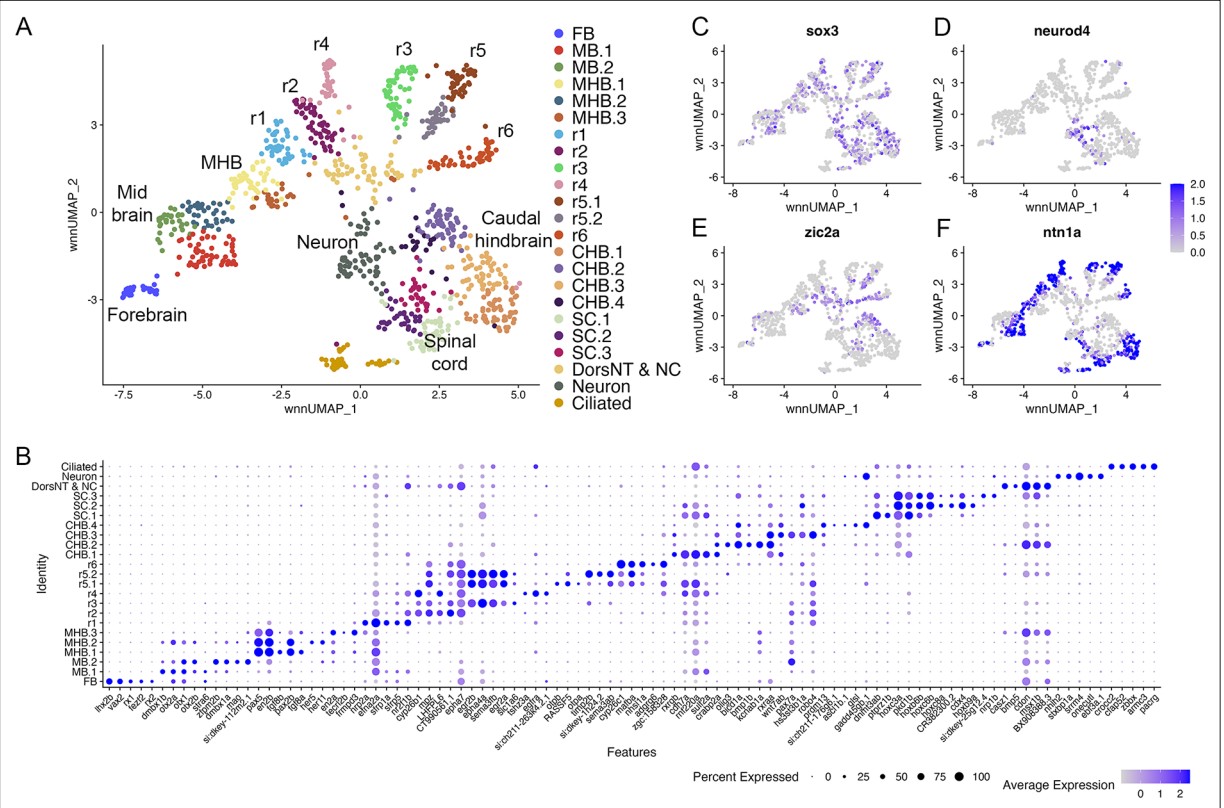

**Figure 2.** Individual rhombomeres are resolved in the 16hpf zebrafish hindbrain. See also *Figure 2—figure supplement 1*. (**A**) UMAP of 16hpf neural clusters. (**B**) Dot plot showing the expression of the top five enriched genes in each cluster. (**C, D**) Feature plots showing the expressing of *sox3* (**C**) and *neurod4* (**D**). (**E, F**) Feature plots showing the expression of dorsoventral marker genes. dorsNT = dorsal neural tube, NC = neural crest. See legend to *Figure 1* for additional abbreviations.

The online version of this article includes the following figure supplement(s) for figure 2:

**Figure supplement 1.** scMultiome analysis of 16hpf zebrafish.

*et al., 1998b*). The r4 cluster is enriched for Pbx binding motifs that are also accessible in some more caudal tissues (*Figure 4G*). Notably, this type of Pbx site is distinct from the heterodimeric TALE sites enriched in r3 and are reported at enhancers where Pbx TFs act as co-factors to Hox proteins – such as at several Hox-dependent enhancers active in r4 (*Ferretti et al., 2000*; *Pöpperl et al., 1995*; *Jacobs et al., 1999*; *Berthelsen et al., 1998*; *Ferretti et al., 2005*; *Tümpel et al., 2007*). Cells in the r2 cluster are highly enriched for accessible binding motifs for TEAD family TFs at 13hpf (*Figure 4H*). TEAD TF expression is not restricted to r2 (*Figure 4A and B*), but TEAD TFs are known heterodimerization partners to Vgll TFs (*Günther et al., 2004*; *Maeda et al., 2002*) and *vgll3* expression is restricted to r2 (*Figure 1J and R–T*). This suggests that Vgll3 supports the binding of TEAD TFs to genomic DNA specifically in r2. We did not identify any TF binding motifs unique to r1, but the r1 and r2 clusters (along with some MHB clusters) display enrichment for NR2C/F family motifs (*Figure 4I and J*), consistent with *nr2f1b* being expressed in this region of the zebrafish embryo (*Love and Prince, 2012*). Lastly, we note that many motifs enriched in the rhombomeres are also enriched in clusters containing differentiating neurons, suggesting that some genomic regions remain accessible as progenitors begin to differentiate. Overall, we find strong agreement between the scRNA and scATAC data in defining individual rhombomeres and we conclude that r1-r6 are molecularly distinct from each other and adjacent neural structures by 13hpf, which precedes the stage when individual rhombomeres become visually detectable in zebrafish.

## Each rhombomere displays a unique molecular profile

The hindbrain possesses a periodicity, such that pairs of one odd- and one even-numbered rhombomere are arranged along its anterior-posterior axis. This concept is primarily based on cellular and

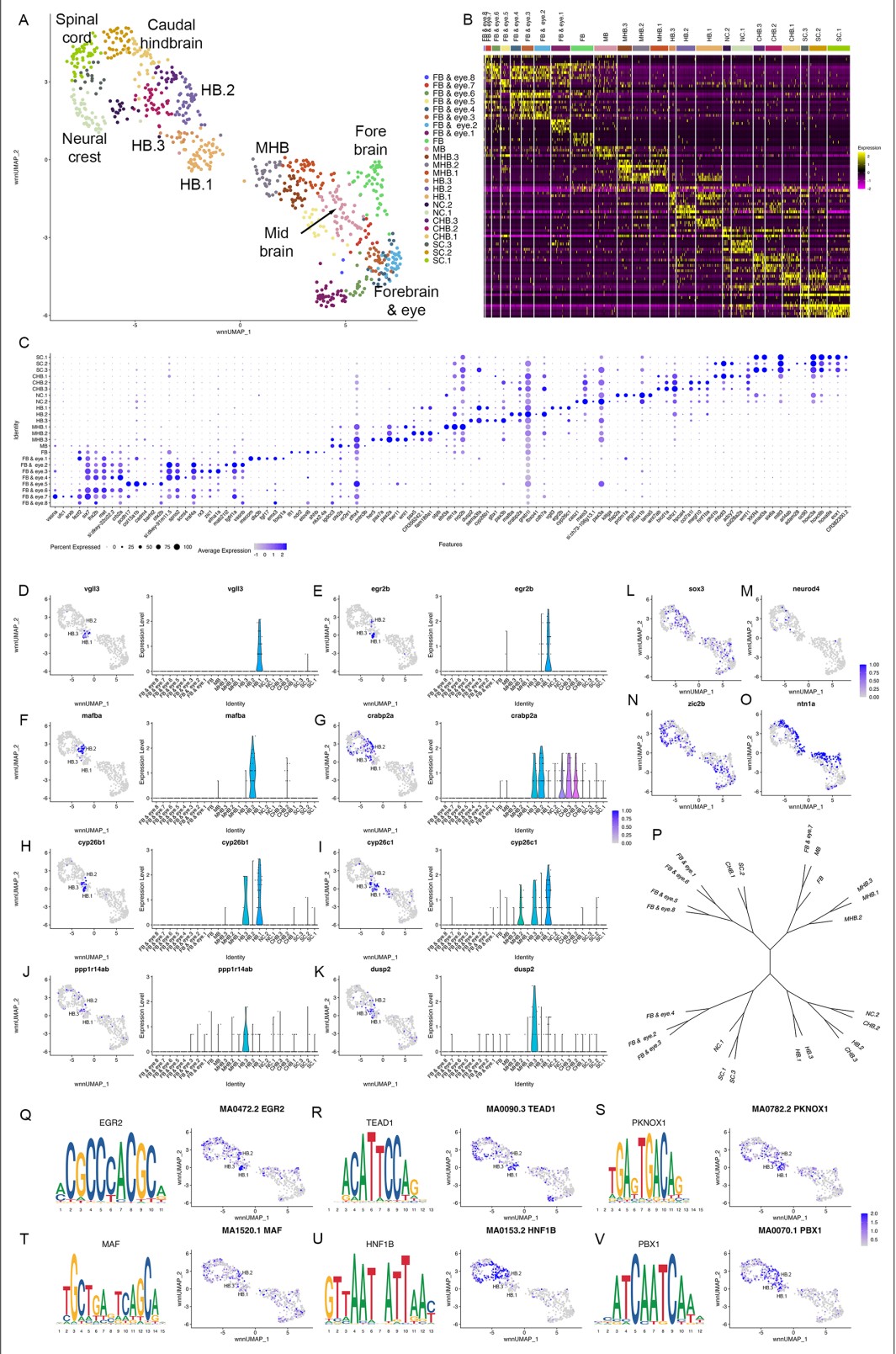

**Figure 3.** Individual rhombomeres are not resolved at 10hpf in zebrafish. See also *Figure 3—figure supplement 1*. (**A**) UMAP of 10hpf neural clusters. (**B**, **C**) Heat map (**B**) and dot plot (**C**) showing the expression of the top five enriched genes in each cluster. (**D-K**) Expression of rhombomere-specific genes are shown as feature plots (left panels) and violin plots (right panels). (**L-M**) Feature plots showing the expressing of *sox3* (**L**) and *neurod4* (**M**). (**N**,

*Figure 3 continued on next page*

*Figure 3 continued*

**O**) Feature plots showing the expression of dorsoventral marker genes. (**P**) Dendrogram showing the relationship between the 10hpf neural clusters. (**Q-V**) Rhombomere-enriched accessible transcription factor binding motifs are shown as a motif logo (left panels) and as a feature plot of chromVar activity (right panel). HB = hindbrain, NC = neural crest. See legend to *Figure 1* for additional abbreviations.

The online version of this article includes the following figure supplement(s) for figure 3:

**Figure supplement 1.** scMultiome analysis of 10hpf zebrafish.

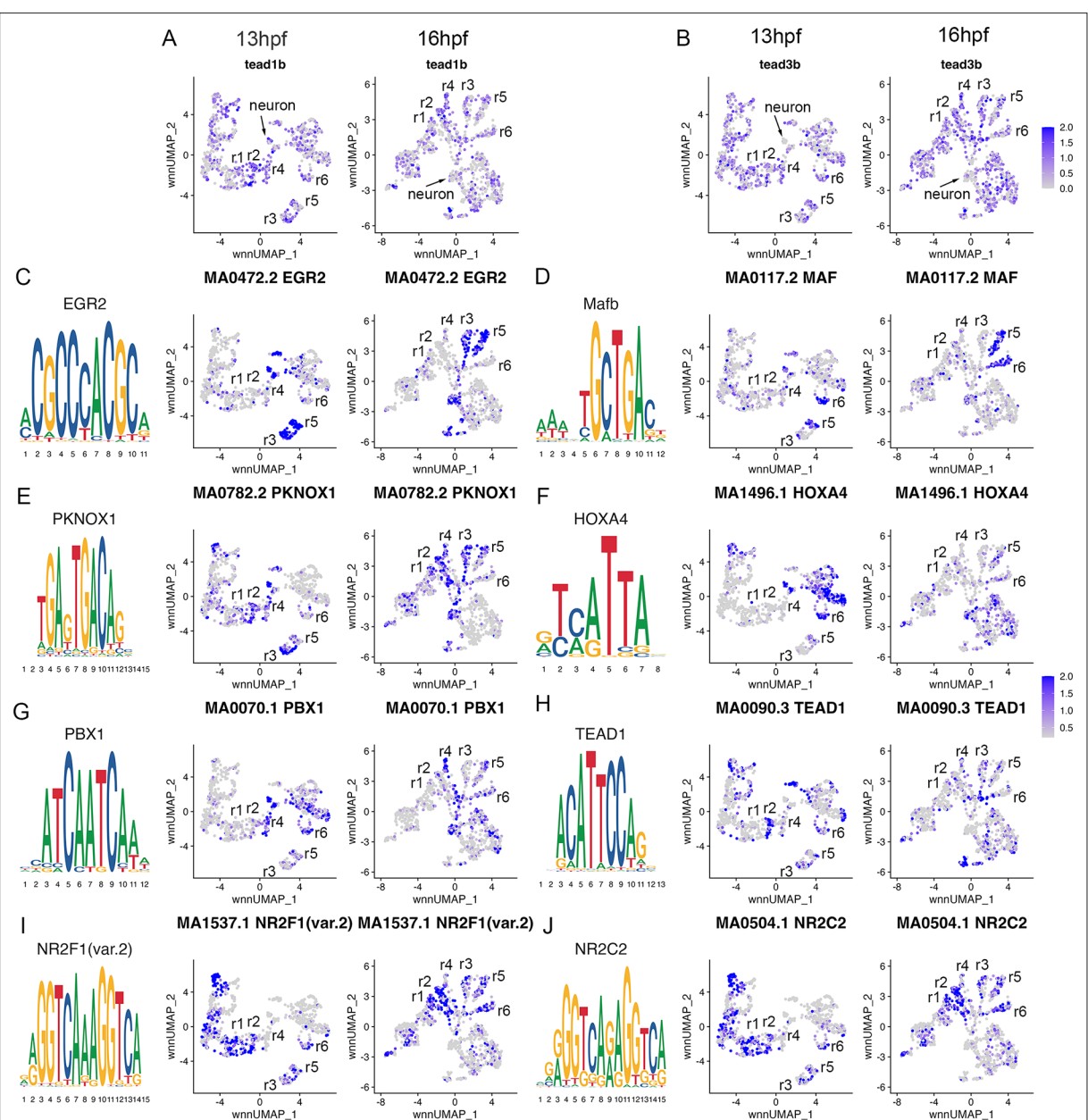

**Figure 4.** Transcription factor binding motifs show rhombomere-restricted accessibility. See also *Figure 4—source data 1*. (**A**, **B**). Feature plots showing the expression of TEAD transcription factors at 13hpf and 16hpf. (**C-J**). Rhombomere-enriched accessible transcription factor motifs are shown as a motif logo (left panels) and as a feature plot of chromVar activity at 13hpf (middle panels), or 16hpf (right panels).

The online version of this article includes the following source data for figure 4:

**Source data 1.** Differentially accessible transcription factor motifs for each cluster at each time point.

anatomical features, such as the exit of branchiomotor nerves, the migration of neural crest cells and cell cycle activity (*Lumsden and Keynes, 1989*; *Lumsden et al., 1991*). Despite the discovery of some genes that recapitulate this pattern (e.g. *epha4a* expression in r1/r3/r5; *Xu et al., 1995*, and *egfl6* expression in r2/r4/r6; *Choe et al., 2011*), it is unclear if this periodicity is present earlier in development and if it is broadly recapitulated molecularly – in the form of gene expression and chromatin accessibility patterns. The UMAPs presented in *Figures 1 and 2* do not show odd and even rhombomeres clustering into separate groups. Instead, r1, r2, and r4 cluster with anterior neural structures, while r6 clusters closer to caudal structures. Since the position of clusters in the UMAP is not fully indicative of how related the cell populations are, we next generated heatmaps of the top genes in each cluster (*Figure 5A, B*). The heatmaps reveal considerable gene expression overlap between r3 and r5, but these rhombomeres do not share extensive expression with r1. Similarly, while r2, r4, and r6 share some gene expression, r2 seems to express more genes related to r3, and r6 shares closer expression with r5. To comprehensively examine these relationships across all rhombomeres, we used the weighted nearest-neighbor graphs from the Seurat analysis and projected them as dendrograms (*Figure 5C, D*). This analysis revealed very similar relationships at 13hpf and 16hpf. At both stages, r3 and r5 are closely related to each other, while r1 is more closely related to the MHB (13hpf) or r2 (16hpf). Furthermore, r4 is related to r2, but not to r6, which instead clusters with the caudal hindbrain at both stages. As a result, there is no clustering of even versus odd rhombomeres at either 13hpf or 16hpf. To more directly assess if there is a molecular odd versus even pattern among rhombomeres, we computationally identified any genes expressed in r2, r4, and r6 – and whose average expression in those rhombomeres exceeds their expression in r1, r3, and r5 – or vice versa at 13hpf (*Figure 5E and F*). We find very few genes that fulfill these criteria. *col7a1l* is highly expressed in r2, r4, and r6, but most other genes expressed in an even or odd patterns are expressed weakly in one of the rhombomeres. For instance, *plxna2* is robustly expressed in r2 and r4, but only weakly in r6, while *egr2b*, *epha4a*, and *sema3fb* are highly expressed in r3 and r5, but barely detectable in r1. We carried out an analogous analysis to identify genes expressed in two adjacent rhombomeres, to determine if rhombomere pairs can be defined by co-expression of genes. This analysis identified several genes co-expressed in r5 and r6 (*Figure 5I*), like the known expression of *mafba* (*Moens et al., 1998*) and the PG3 and PG4 *hox* genes (*Prince et al., 1998a*; *Prince et al., 1998b*). In contrast, we detect only a few genes co-expressed in r1 and r2 (*nrp2a* and *pdzrn4*; *Figure 5G*) and none co-expressed in r3 and r4. We next used the Direct-Net tool (*Zhang et al., 2022*) to delineate and compare the gene regulatory networks (GRNs) of 13hpf rhombomeres (*Figure 5—figure supplements 1–7*). This analysis revealed that each rhombomere is characterized by a unique GRN with previously known, as well as novel, TFs acting as nodes. For example, the TEAD1 TF forms a predominant node in r2 (*Figure 5—figure supplement 2*), as expected based on its known interaction with Vgll3 that is uniquely expressed in r2, but so do the Sox6 and Esrrg TFs that have not been previously implicated in r2 formation. While some TFs are shared between rhombomeres, such as Nr2f family TFs forming nodes in both the r1 and r2 GRNs (*Figure 5—figure supplements 1 and 2*), the GRNs are clearly distinct. Even when comparing rhombomeres that share significant molecular similarity – such as r3 and r5 that both have a predominant Egr2 node – we find that their GRNs also possess numerous distinct nodes – such as the PG 3 and 4 *hox* genes in r5 but not r3 (*Figure 5—figure supplements 3 and 5*). Lastly, even TFs that are shared between rhombomeres are connected to different genes in each GRN, as illustrated by Egr2 connecting to MafB and RARa in r5, but not r3 (*Figure 5—figure supplement 7*). This analysis represents the first comprehensive rhombomere-specific GRNs and extends prior work that generated GRNs based on bulk data, or incompletely separated rhombomere clusters (*Tambalo et al., 2020*; *Krumlauf and Wilkinson, 2021*). We conclude that each 13hpf rhombomere is characterized by a unique GRN that defines a specific molecular identity, and that there is not an overt molecular odd versus even, or rhombomere pair, pattern at this stage.

## scMultiome identifies three hindbrain progenitor domains at the end of gastrulation

To determine when rhombomeres emerge during embryogenesis, we examined the 10hpf neural UMAP. Using gene expression (*Figure 1—source data 2*) and motif accessibility (*Figure 4—source data 1*), we assigned cell identities to each cluster (*Figure 3A–C*) and noted that dorsoventral patterning is already ongoing at this stage (*Figure 3N and O*). We find that, in contrast to 13hpf

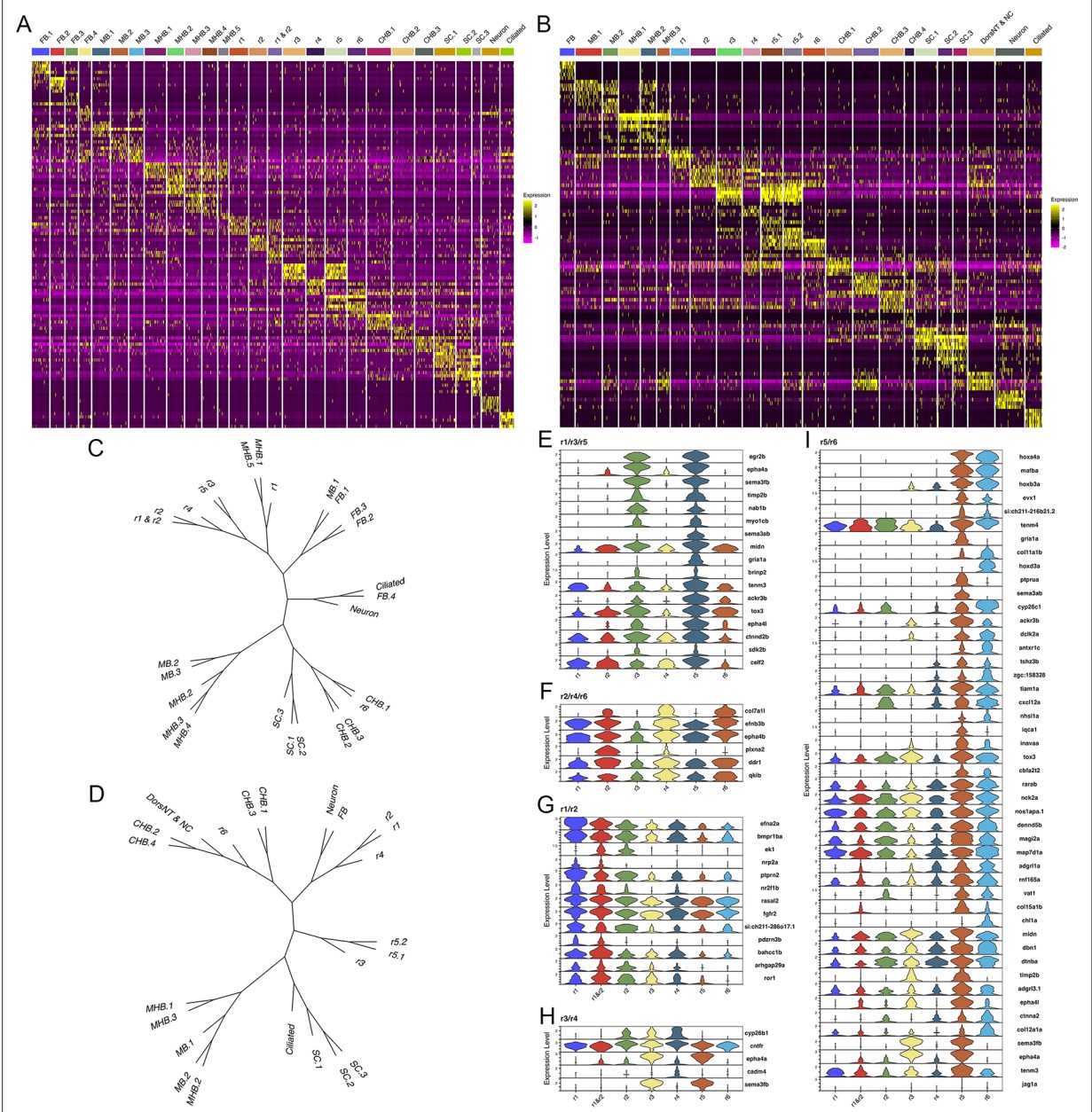

**Figure 5.** Each rhombomere displays a unique molecular profile. See also *Figure 5—figure supplements 1–7*. (**A**, **B**) Heat maps showing the expression of the top five enriched genes in each cluster at 13hpf (**A**) and 16hpf (**B**). Gene order in A and B is the same as in *Figures 1B and 2B*, respectively. (**C**, **D**). Dendrograms showing the relationship between each neural cluster at 13hpf (**C**) and 16hpf (**D**). ( **E**-**I**). Violin plots showing expression levels of genes enriched in odd (**E**) or even (**F**) rhombomeres, as well as genes enriched in adjacent pairs of rhombomeres (**G**–**I**), at 13hpf.

The online version of this article includes the following figure supplement(s) for figure 5:

**Figure supplement 1.** Direct-Net derived gene regulatory network for rhombomere 1 at 13hpf.

**Figure supplement 2.** Direct-Net derived gene regulatory network for rhombomere 2 at 13hpf.

**Figure supplement 3.** Direct-Net derived gene regulatory network for rhombomere 3 at 13hpf.

**Figure supplement 4.** Direct-Net derived gene regulatory network for rhombomere 4 at 13hpf.

**Figure supplement 5.** Direct-Net derived gene regulatory network for rhombomere 5 at 13hpf.

**Figure supplement 6.** Direct-Net derived gene regulatory network for rhombomere 6 at 13hpf.

**Figure supplement 7.** Direct-Net derived Egr2 networks for rhombomeres 3 and 5 at 13hpf.

and 16hpf, individual rhombomeres are not apparent as distinct clusters at 10hpf. Instead, we identified three clusters (HB.1-HB.3) that express genes characteristic of mature rhombomeres. HB.1 is the largest of these and expresses genes associated with several different rhombomeres such that *vgll3* (*Figure 3D*; an r2 marker; *Figure 1R*; *Simon et al., 2017*) and *egr2b* (*Figure 3E*; a marker of r3 and r5; *Oxtoby and Jowett, 1993*) are both enriched in this cluster. A second prominent rhombomere cluster (HB.2) expresses *mafba* (*Figure 3F*; a marker of r5/r6; *Moens et al., 1998*) and *crabp2a* (*Figure 3G*; expressed in r6, the caudal hindbrain, and weakly in r4/r5; *Sharma et al., 2005*), as well as limited *egr2b* (*Figure 3E*). Since *egr2b* expression is initiated in r3 at the end of gastrulation (10hpf), but does not become expressed in r5 until 11hpf, this suggests that HB.1 and HB.2 correspond to prospective anterior and posterior rhombomeres, respectively. Furthermore, a third rhombomere cluster (HB.3) is related to HB.1 as these two clusters co-express *cyp26b1* and *cyp26c1* (*Figure 3H, I*; expressed in r2-r4; *Zhao et al., 2005*; *Gu et al., 2005*), but HB.3 does not express either *vgll3* or *egr2b* (*Figure 3D and E*). Instead, HB.3 expresses *dusp2* and *ppp1r14ab*, two markers that we previously reported as restricted to r4 at this stage (*Figure 3J and K*; *Choe et al., 2011*; *Maurer and Sagerström, 2018*). All HB clusters are clearly separate from a group of clusters corresponding to the MHB, the midbrain and the forebrain, while the HB.2 cluster is adjacent to several clusters corresponding to the caudal hindbrain and the anterior spinal cord. TF motif accessibility generally supports the gene expression analysis, and we find that multiple TF binding motifs are associated with the HB clusters at 10hpf. Egr, TEAD, and TALE binding motifs are enriched in accessible chromatin regions in HB.1 cluster cells (*Figure 3Q-S*), which is supportive of our assertion that HB.1 contains precursors of anterior rhombomeres. Similarly, the HB.2 cluster is enriched for accessible Maf and Hnf1 motifs in accordance with it containing posterior progenitors (*Figure 3T and U*). Egr motifs are also weakly enriched in HB.2, consistent with the *egr2b* gene becoming activated in r5 at this stage. Lastly, we find that Pbx motifs are enriched in HB.3 (*Figure 3V*). Although Pbx motifs are also enriched in HB.1 and HB.2, HB.3 is unique in that it shows minimal enrichment for accessible Egr, TEAD, Hnf1, and Maf motifs. These relationships are supported by dendrogram projections (*Figure 3P*), where HB.2 clusters with the caudal hindbrain, whereas HB.1 and HB.3 cluster together on a separate branch. Taken together, this analysis demonstrates that individual rhombomeres are not yet molecularly distinct at 10hpf. Instead, three progenitor domains with different anteroposterior characteristics are present, such that HB.1 likely corresponds to the future r2/r3 (and possibly r1), HB.2 to r5/r6 and HB.3 to r4. We tentatively refer to these domains as primary hindbrain progenitor domains (PHPDs).

## The PHPDs contain progenitors with mixed rhombomere identities

To determine how the PHPDs relate to rhombomeres, we integrated neural clusters from all three timepoints. To simplify this analysis, we omitted all forebrain cells and projected the remaining cells as a UMAP (*Figure 6A*). The integrated UMAP reveals clearly delineated clusters, and we identified each rhombomere based on its gene expression and TF motif accessibility profile (*Figure 6—source data 1 and 2*). We again find that each rhombomere is split into dorsal (expressing *zic2b*) and ventral (marked by *ntn1a*) clusters – except for r3, which is represented by a single cluster that nevertheless shows differential expression of dorsoventral genes in distinct subdomains (*Figure 6B, C*). Projecting the integrated data in the form of a dendrogram (*Figure 6—figure supplement 1A*) largely confirms the relationships observed at individual timepoints such that r1 and r2 cluster together with r4 on a shared branch, r3 and r5 cluster on a separate branch, and r6 forms its own branch. We then used the integrated UMAP to examine the contribution of cells from each timepoint to the integrated rhombomere clusters (*Figure 6D*; *Figure 6—source data 3*). We find that cells from individual rhombomeres at 13hpf and 16hpf combine into integrated clusters such that, for instance, r3 cells from 13hpf and 16hpf contribute to the integrated r3 cluster, indicating that hindbrain cells have shared rhombomere characteristics at 13hpf and 16hpf. In contrast, cells from the 10hpf PHPD clusters (HB.1–3) do not contribute fully to the integrated rhombomeres. A portion of cells from the HB.2 cluster distribute to the integrated r5 and r6 clusters, but many remain as a separate cluster (HB) in the integrated UMAP (arrow in *Figure 6D*), indicating that – although they express *mafba* (a marker of r5/r6; *Figure 3F*) – their overall molecular profile is sufficiently distinct from r5/r6 cells that they remain as a separate group. Even the HB.2 cells that do distribute to rhombomeres appear located between r4 and r6 in the integrated UMAP. Similarly, while cells from the HB.1 cluster distribute to the integrated r1, r2, and r3 clusters, a detailed examination of their distribution reveals that many

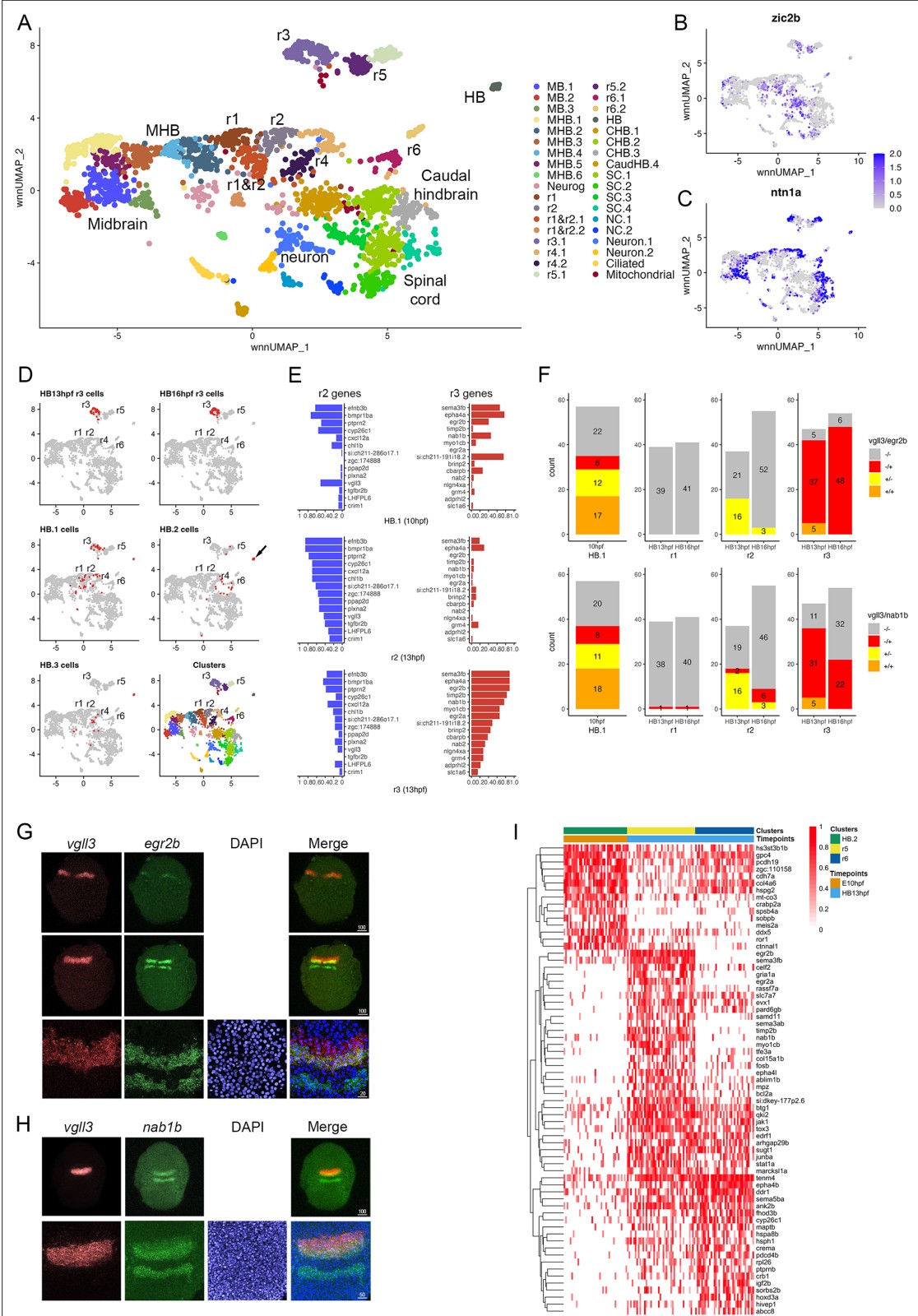

**Figure 6.** Rhombomere progenitor cells display mixed gene expression identities. See also *Figure 6—figure supplement 1*, *Figure 6—source data 1*, *Figure 6—source data 2* and *Figure 6—source data 3*. (**A**). UMAP of integrated data from 10hpf, 13hpf and 16hpf. (**B**, **C**). Feature plots showing the expression of dorsoventral marker genes. (**D**). Feature plots showing the contribution of various cell populations (listed at the top of each panel and shown as red cells) to the integrated rhombomere clusters. (**E**). Bar graphs showing expression of r2 genes (left column) and r3 genes (right column)

*Figure 6 continued on next page*

*Figure 6 continued*

in 10hpf HB.1 cells (top panel), 13hpf r2 cells (middle panel) and 13hpf r3 cells (bottom panel). (**F**). Co-expression of r2 markers (*vgll3*) and r3 markers (*egr2b, nab1b*) in individual cells from HB.1, r1, r2, and r3. Numbers in the bar graphs represent cell counts. (**G**, **H**). HCR analysis of rhombomere-restricted gene expression in 10-11hpf wild-type zebrafish embryos. Embryos are shown in dorsal view with anterior to the top. The bottom row in each panel displays a higher magnification of the row above. (**I**). Heatmap showing genes differentially expressed between HB.2 at 10hpf and its derivative rhombomeres (r5 and r6) at 13hpf. Scale bar sizes in (**G**, **H**) are given in um.

The online version of this article includes the following source data and figure supplement(s) for figure 6:

**Source data 1.** Differential gene expression data for each cluster in the integrated data set.

**Source data 2.** Differentially accessible transcription factor motifs for each cluster in the integrated dataset.

**Source data 3.** Origin of cells contributing to each rhombomere cluster in the integrated UMAP.

**Figure supplement 1.** Characterization of gene expression in primary hindbrain progenitor domains (PHPDs) relative to their derivative rhombomeres.

occupy a position between r1 and r2, further indicating that PHPD cells are molecularly distinct from rhombomere cells.

To better characterize the transcriptional state of PHPD cells, we examined the HB.1 cluster more closely. We identified the top genes differentially expressed in r2 versus r3 at 13hpf (and vice versa). As expected, these genes are preferentially expressed in their corresponding rhombomere at 13hpf (*Figure 6E*, middle and bottom panels), but we find that both r2 and r3 genes are extensively expressed in HB.1 cells at 10hpf (*Figure 6E*, top panel). Examining the expression state of individual HB.1 cells, we identified many cells that express both *vgll3* (r2 marker) and *egr2b* (r3 marker at this stage), such that ~74% of *egr2b*-expressing cells co-express *vgll3* at 10hpf (*Figure 6F*; top left panel). This contrasts with 13hpf – when r2 cells do not co-express *egr2b* and only ~10% of r3 cells co-express *vgll3* – and 16hpf, when *egr2b* is fully restricted to r3 and *vgll3* is being downregulated (*Figure 6F*; top right panels). In addition to co-expressing *egr2b*, we find that *vgll3*-positive HB.1 cells also co-express *nab1b* (r3 and r5 marker; *Mechta-Grigoriou et al., 2000*; *Figure 1T*) such that 62% of *vgll3* positive HB.1 cells co-express *nab1b* (*Figure 6F*; bottom left panel). When considered in combination, ~72% of *vgll3*-positive HB.1 cells co-express either *egr2b* or *nab1b* (*Figure 6—figure supplement 1B*). To identify the HB.1 domain and assess the co-expression of rhombomere markers in vivo, we turned to HCR analysis in zebrafish embryos. We find extensive overlap between *vgll3* and both *egr2b* and *nab1b* expression at the earliest stages that these genes can be detected (*Figure 6G and H*), indicating that HB.1 cells possess mixed rhombomere identities in vivo.

To begin characterizing the transition from PHPD cells to rhombomere cells, we made use of our time course of scMultiome data. We compared the molecular profile of cells in each PHPD at 10hpf to that of its derivative rhombomere cells at 13hpf and projected as heatmaps (*Figure 6I*; *Figure 6—figure supplement 1C*). This analysis detected the upregulation of genes known to become enriched in rhombomeres, such as *egr2b* in r5, *hoxd3a* in r6, and *hoxa4a* in r5/r6 (*Figure 6—figure supplement 1D–F*), as well as some genes not previously implicated in rhombomere formation (e.g. *cxcl12a*; *Figure 6—figure supplement 1G*). We also identified genes that are expressed more highly in the PHPDs and that may need to be downregulated for the transition to specific rhombomere identities (e.g. *ror1*; *Figure 6—figure supplement 1H*). We conclude that the PHPDs differ from later rhombomeres in terms of their molecular profile and, at least in the case of HB.1, individual PHPD cells possess mixed identities characterized by co-expression of markers for multiple rhombomeres. During the development of PHPD cells into rhombomeres, their gene expression profiles undergo a corresponding change.

## Morphogens control formation of the PHPDs

The anteroposterior pattern of the early hindbrain primordium is established by morphogens (reviewed in *Krumlauf and Wilkinson, 2021*; *Frank and Sela-Donenfeld, 2019*), with RA acting posteriorly (from its source in the adjacent somites) and Fgfs acting anteriorly (with an initial source at the MHB followed by a secondary source in r4). Partial disruption of RA signaling has variable effects on hindbrain gene expression (reviewed in *Begemann and Meyer, 2001*), whereas complete inhibition appears to block the formation of caudal rhombomeres (*Dupé and Lumsden, 2001*), suggesting that RA may be required for the formation of the HB.2 domain. Accordingly, we find that expression of RA signaling components – such as the *RARab* retinoic acid receptor and the *crabp2a* retinoic

acid binding protein – is enriched in HB.2 cells (*Figure 7A and B*). To test if formation of the HB.2 domain requires RA signaling, we used DEAB (an inhibitor of RA synthesis) to block RA signaling and assessed early HB.2 gene expression. We find that *egr2b* expression in r5 and *mafba* expression in r5/r6 is detected by 10hpf-11hpf in control embryos but is never observed in DEAB-treated embryos (*Figure 7H–K*). In contrast, the expression of the anterior markers (*vgll3*, *cyp26b1*, *cyp26c1*, and the anterior *egr2b* domain) persists in DEAB-treated embryos, but the expression is observed over a broader region and may be somewhat weaker. Previous work demonstrated that disruption of r5/r6 formation in *mafB* mutant embryos results in a unique rX cell population between r4 and the caudal hindbrain (*Moens et al., 1998*; *Moens et al., 1996*). To determine if such an rX domain forms in place of r5/r6 in DEAB-treated embryos, we assessed expression of *cdx4* in the spinal cord relative to *cyp26b1* in r4 and *egr2b* in r3 (*Figure 7L and M*). We again observe a slight expansion of *egr2b* in r3 and *cyp26b1a* in r4 of DEAB-treated embryos, but these domains closely abut *cdx4* expression. Inhibition of RA signaling, therefore, does not appear to induce an rX domain, but leads to a loss of the caudal hindbrain and an expansion of r4, consistent with a role for RA in HB.2 formation. Since each rhombomere produces a specific set of neuronal progenitors, we next examined if the loss of HB.2 leads to corresponding loss of differentiated neurons (*Figure 7Q–R*). We find that vagal (nX) neurons in the caudal hindbrain and abducens (nVI) neurons in r5/r6 are largely absent in DEAB-treated embryos while r4-derived facial (nVII) neurons and r4-specific reticulospinal Mauthner neurons (MNs) appear expanded (2 MNs per wild-type embryo versus 3.6 MNs per DEAB-treated embryo). We conclude that loss of RA signaling leads to loss of the HB.2 domain and to the expansion of the HB.3 domain.

In agreement with a role also for Fgf signaling, we detect expression of Fgf8 and Fgf3 in several PHPD clusters of the 10hpf UMAP (*Figure 7C and D*), and we find that treatment with SU5402 (an inhibitor of Fgf receptor signaling) blocks expression of *vgll3* and *egr2b* in r2/r3 at 10hpf (*Figure 7N*) – as well as of *mafba* and *egr2b* in r5/r6 and of *hoxb1a* in r4 – when those genes first become detectable at 12hpf-13hpf (*Figure 7O and P*). Therefore, while RA is required for HB.2 formation, Fgf appears broadly required to establish the PHPDs.

While these analyses confirm a role for morphogens in patterning the hindbrain primordium and suggest that RA and Fgf are required for PHPD formation, they rely on HCR analyses of a small number of genes. It is, therefore, possible that the expression of other rhombomere markers persists and/or that the remaining rhombomeres are misspecified in embryos with disrupted morphogen signals. Previous analyses of morphogen function in early hindbrain development suffers from the same imitations, as they also did not comprehensively assess the effects on rhombomere gene expression. To address this shortcoming, we again turned to scMultiome analysis. Since RA signaling appears to be required for HB.2 formation we collected DEAB-treated embryos at 13hpf (*Figure 7—figure supplement 1*), and combined DEAB-treated with control samples into an integrated UMAP (*Figure 7E*). In assessing the contribution of control versus DEAB-treated cells to each UMAP cluster, we find that the r5 and r6 clusters contain only cells from control embryos and no cells from DEAB-treated embryos (*Figure 7F*). Accordingly, projecting DEAB-treated and wild-type samples in separate UMAPs reveals a complete loss of r5 and r6 in the DEAB condition (*Figure 7G*), consistent with the HB.2 progenitor population not forming. In agreement with the apparent expansion of anterior rhombomeres observed in our HCR analyses of DEAB-treated embryos, cell ratio comparisons reveal that r1 and r2 contain twice as many cells, and r4 contains almost three times as many cells, in DEAB-treated compared to control embryos (*Figure 7F*). Furthermore, the caudal hindbrain (CHB) clusters are lost in DEAB-treated embryos while the spinal cord clusters largely persist (*Figure 7F and G*). We also find that all cells from DEAB-treated embryos can be mapped to a wild-type cluster, again indicating that an rX domain does not form in DEAB-treated embryos. Furthermore, we find only minor changes in gene expression in r1-r4 upon loss of RA signaling (*Figure 7—figure supplement 2*; *Figure 7—source data 1*). Only ~30 genes are differentially expressed between DEAB-treated and control embryos (>twofold change; p-val <1e-5) in each rhombomere – and many of these genes are shared between rhombomeres – indicating that they reflect broadly RA-responsive genes rather than genes that affect individual rhombomere fates. Our results suggest that RA is required for formation of the HB.2, but not the HB.1 or HB.3 domains, although we cannot exclude the possibility that a small number of genes are RA-dependent in the anterior domains.

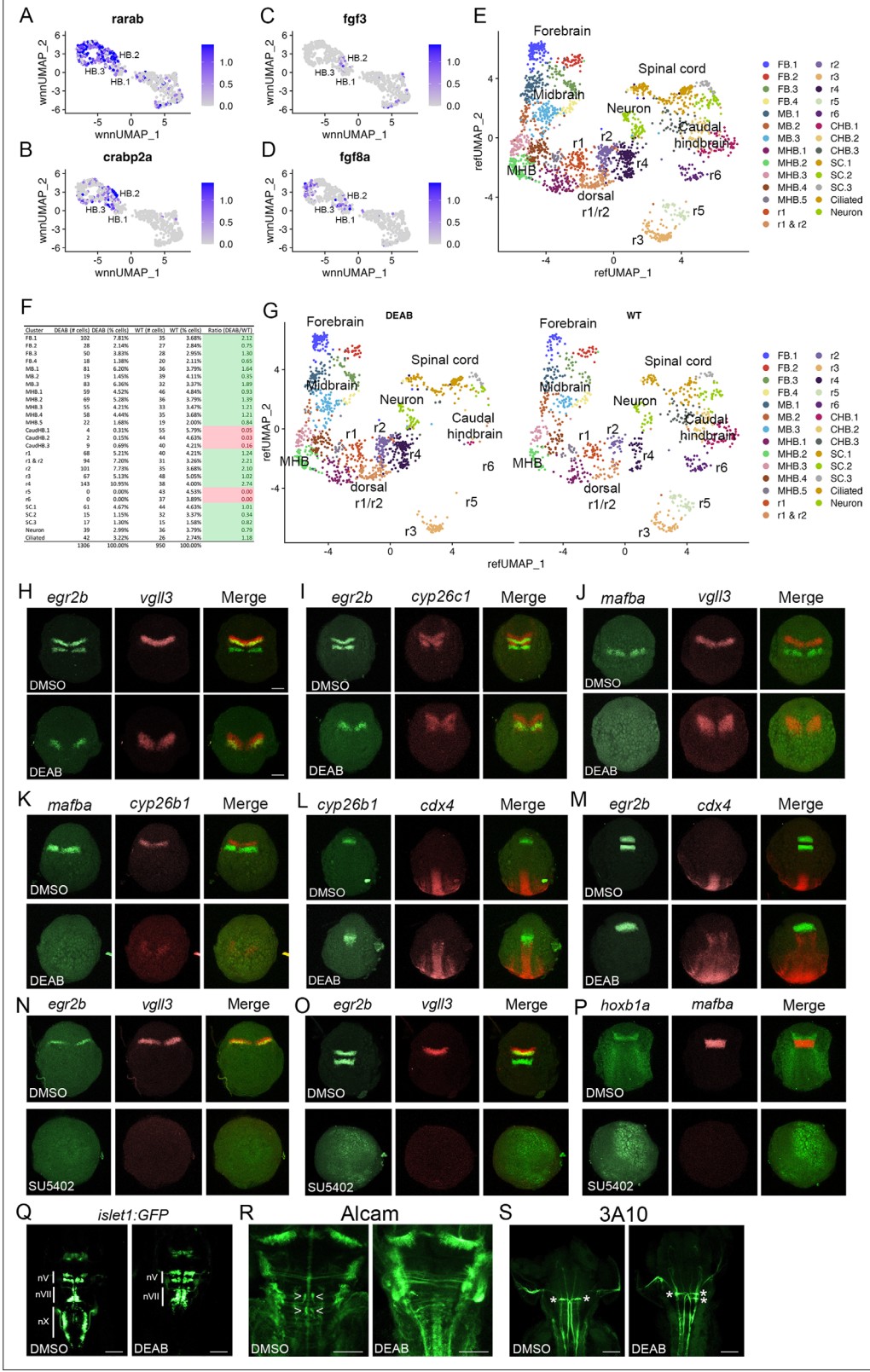

**Figure 7.** Primary hindbrain progenitor domain (PHPD) formation is controlled by morphogens. (**A**-**D**). Feature plots showing the expression of retinoic acid (RA) and fibroblast growth factor (Fgf) signaling components at 10hpf. (**E**). Combined UMAP of 13hpf wild-type and DEAB-treated embryos. (**F**). Contribution of wild-type (WT) and DEAB-treated cells to each cluster in (**E**). (**G**). Separate UMAP projections of wild-type (right panel) and DEAB-

*Figure 7 continued on next page*

*Figure 7 continued*

treated (left panel) embryos at 13hpf. (**H-M**). Hybridization chain reaction (HCR) analysis of rhombomere-restricted gene expression in control (top panels) and DEAB-treated (bottom panels) embryos at 10-11hpf. N-P. HCR analysis of rhombomere-restricted gene expression in control (top panels) and SU5402-treated (bottom panels) embryos at 10-11hpf (**N**) or 12-13hpf (**O–P**). HCR marker gene expression: *egr2b*: r3/5; *vgll3*: r2; *cyp26c1*: r2/4; *mafba*: r5/r6; *cyp26b1*: r4; *hoxb1a*: r4; *cdx4*: spinal cord. (**Q**). Detection of branchiomotor neurons in DMSO- (left panel) and DEAB- (right panel) treated *TG(isl1a:GFP^rw0)* transgenic embryos at 48hpf. (**R**, **S**). Immunostaining for abducens (Alcama antibody) and Mauthner (3A10 antibody) neurons in DMSO- (left panel) and DEAB- (right panel) treated embryos at 48hpf. Embryos are shown in dorsal view with anterior to the top. Scale bar in H is for panels (**H-P**) (100 um). Scale bars in (**Q-S**) are 100 um.

The online version of this article includes the following source data and figure supplement(s) for figure 7:

**Source data 1.** Differential gene expression between control and DEAB-treated rhombomeres 1–4 at 13hpf.

**Figure supplement 1.** scMultiome analysis of DEAB-treated 13hpf zebrafish.

**Figure supplement 2.** DEAB treatment has a limited effect on gene expression in rhombomeres 1–4.

## Discussion

Hindbrain rhombomeres represent a well-studied model for neural progenitor specification and positioning, but we still lack a clear understanding of the molecular basis for differences between rhombomeres, and we do not know how they form in the embryo. This is largely due to a lack of comprehensive molecular data as individual rhombomeres have not been resolved by previous scRNAseq analyses on whole embryos (*Jiang et al., 2021*; *Farnsworth et al., 2020*; *Wagner et al., 2018*; *Farrell et al., 2018*), or on dissected hindbrain regions (*Tambalo et al., 2020*). Using combined single nucleus RNAseq and ATACseq (scMultiome), we successfully resolved all rhombomeres at zebrafish segmentation stages. We identify a unique molecular profile for each rhombomere and find that there is no overt odd versus even periodicity of rhombomere identities at these stages. We also define three mixed-identity progenitor domains that likely correspond to the pre-rhombomeres observed visually in early embryos of several species and that are predicted to subsequently subdivide into the mature rhombomeres. Our results provide a comprehensive molecular basis for the formation and unique identities of hindbrain rhombomeres.

### Early rhombomeres lack an overt molecular periodicity

The embryonic hindbrain possesses an odd versus even periodicity of cellular and anatomical features (*Lumsden and Keynes, 1989*; *Lumsden et al., 1991*), but the molecular basis of this pattern is not clear, and it is not known how early it may be present. Using the full gene expression profile for each rhombomere at segmentation stages, we find that r3 and r5 are closely related and that r2 shares features with r4, but r1 and r6 do not fall into these groups (*Figure 5*). Indeed, we find very few genes that are restricted to odd versus even rhombomeres at these stages. Similarly, while we observe numerous genes co-expressed in r5 and r6, we do not detect shared gene expression in other pairwise combinations of rhombomeres. Instead of a molecular periodicity, each rhombomere displays a distinct molecular profile that can be captured in a GRN (*Figure 5—figure supplements 1–7*). The GRNs of individual rhombomeres are unique – even closely related rhombomeres (e.g. r3 and r5) have clearly distinct GRNs. Our analyses of gene expression, TF motif accessibility, and GRN formation combine to indicate that repeating two-segment identities are not molecularly evident in the early zebrafish hindbrain.

Although the rhombomeres do not display an overt molecular periodicity in our analyses, previous work demonstrated odd versus even rhombomere features. For instance, transplantation and cell mixing experiments revealed that cells from odd rhombomeres mix more readily into other odd rhombomere than into even ones, and vice versa (*Guthrie and Lumsden, 1991*) – a process likely controlled by cell surface molecules such as the *ephrins* and their *Eph* receptors (*Xu et al., 1995*; *Xu et al., 1999*; *Cooke et al., 2001*). Accordingly, we find that *epha7* is part of the Egr2 regulatory network in both r3 and r5 (*Figure 5—figure supplement 7*). If cell surface receptors are expressed in an even versus odd pattern to drive cell sorting, why do we not detect a corresponding molecular periodicity? The answer may be that these shared expression modules are a minor component of the overall GRN in each rhombomere. Indeed, prior work also revealed a hierarchy for cell mixing

such that, for instance, r3 cells mix more readily with other r3 cells than with cells from different odd-numbered rhombomeres (*Guthrie et al., 1993*) – indicating that the GRNs controlling expression of cell surface receptors may not be the same in all odd rhombomeres. In agreement, we find that *epha2* is part of the Egr2 network in r5, but not in r3 (*Figure 5—figure supplement 7*). In fact, if we compare the Egr2 regulatory network in r3 versus r5 (the two most closely related rhombomeres where Egr2 is an essential TF), we find that only 37% of nodes are shared. Hence, while some GRN components are shared between rhombomeres, these appear to be a minority and, in most cases, may not give rise to detectable molecular periodicities when we compare rhombomeres.

## Three molecularly defined progenitor domains may correspond to pre-rhombomeres

It remains unclear how rhombomeres arise during embryogenesis. Although the hindbrain primordium – together with the entire neural tube – becomes patterned along its AP axis via the action of various morphogens (e.g. fibroblast growth factors, retinoic acid, wnt proteins, etc.; reviewed in *Moens and Prince, 2002*), visual observations reveal that the rhombomeres do not form in a strict AP order. In zebrafish, the r3/r4 and r4/r5 borders form first, which delineates r4 as the first rhombomere. This is followed by the r1/r2 boundary, which leaves a transient r2 + r3 structure that is subsequently subdivided by the formation of the r2/r3 boundary. Similarly, a transient r5 + r6 segment is formed before being subdivided by the r5/r6 boundary (*Moens et al., 1998*; *Moens and Prince, 2002*). The formation of early transient segments may be evolutionarily conserved, since three potentially analogous 'pre-rhombomeres' have been observed in chick and four 'primary rhombomeres' (where the fourth corresponds to the caudal hindbrain) are present in human embryos (*Vaage, 1969*; *Müller and O'Rahilly, 1997*; *Müller and O'Rahilly, 2003*). Furthermore, these early structures appear not to be solely a morphological phenomenon, but to have a genetic basis. In zebrafish *valentino* mutants (that lack *mafb* TF function), r5 and r6 do not form properly, but are replaced by an 'rX' rhombomere that is postulated to contain precursors for r5 + r6 (*Moens et al., 1998*; *Moens et al., 1996*). rX has not been well defined and it is not clear if it is present in wild-type embryos, nor if there are analogous progenitors for the r2 + r3 segment. Our 10hpf Multiome data identify three hindbrain populations (PHPDs),

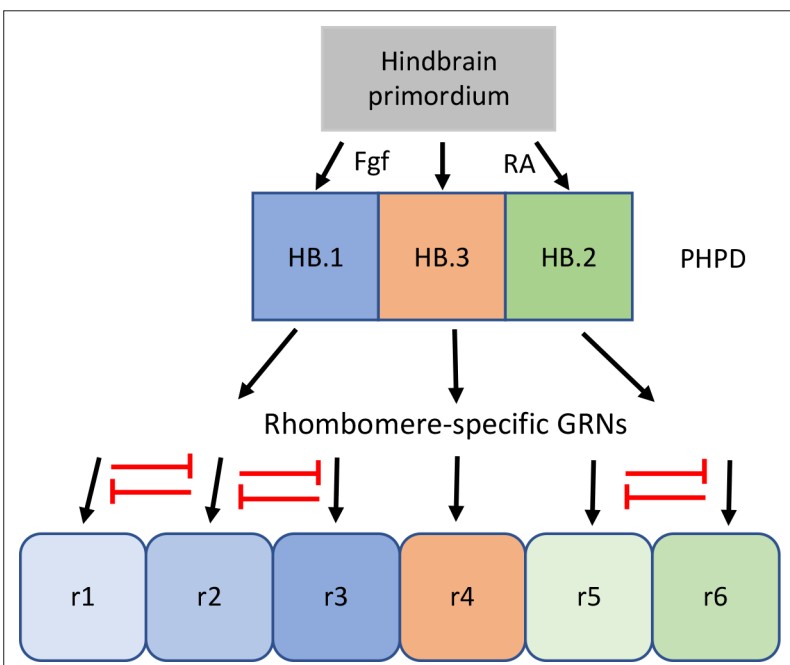

**Figure 8.** Proposed model for the transition from mixed identity primary hindbrain progenitor domain (PHPD) cells to rhombomere cells. An early ground state that lacks an anteroposterior pattern is proposed to respond to morphogens (fibroblast growth factor (Fgf) anteriorly and retinoic acid posteriorly) to generate three PHPDs. Each PHPD contains progenitor cells with mixed identities (HB.1=r1, r2, r3; HB.2=r5, r6, HB.3=r4) that are subsequently resolved via repressive TF interactions (red lines) into rhombomere cells with a single transcriptional identity.

and our analyses demonstrate that they contain rhombomere progenitors. The HB.2 population contains r5 + r6 progenitors, suggesting that it corresponds to rX, while HB.3 appears to correspond to the r4 segment, and HB.1 to the r2 + r3 segment. Based on *Figure 6D*, HB.1 may also contain r1 progenitors, but we cannot confirm this as we do not have an r1 marker at 10hpf. These analyses provide the first molecular characterization of the progenitor domains that precede the formation of individual rhombomeres. It is likely that the PHPDs in turn arise from an earlier unpatterned primordium. Indeed, analyses by several groups have shown that blockade of all Hox TF function – by inhibiting key Hox cofactors – produces a structure that uniformly expresses early hindbrain markers (*Waskiewicz et al., 2002*; *Choe and Sagerström, 2005*). It remains unclear if an equivalent structure exists in wild-type embryos, but scMultiome analyses at earlier stages of embryogenesis might allow a molecular characterization of such a primordium.

## PHPD cells have mixed gene expression profiles that must be resolved during rhombomere formation

Our data reveal that the HB.1 cluster expresses markers of both r2 (*vgll3*) and r3 (*egr2b, nab1b*) at 10hpf (*Figure 3*). A closer examination further demonstrated that individual HB.1 cells co-express r2 and r3 markers and such co-expression was confirmed by HCR analyses in developing embryos (*Figure 6*). These findings indicate that HB.1 cells possess a mixed gene expression profile and may be able to take on both r2 and r3 fates. Accordingly, when 10hpf cells are computationally integrated with cells from 13hpf and 16hpf, they contribute to both the r2 and r3 clusters (*Figure 6*). Notably, the gene expression profiles of HB.1–3 cells are distinct from those of their corresponding rhombomere derivatives and our comparisons between developmental timepoints identified a profound change in gene expression as PHPD cells transition to rhombomere cells (*Figure 6* and *Figure 6—figure supplement 1*). This indicates that the mixed gene expression profiles are resolved during rhombomere formation (*Figure 8*). Analogous to dorsoventral patterning of the neural tube by repressive interactions between TFs (*Briscoe and Novitch, 2008*; *Kutejova et al., 2016*), we hypothesize that TFs expressed in the PHPDs repress each other to drive distinct rhombomere fates. For example, *vgll3* and *egr2b* (or components of their respective GRNs) may repress each other's transcription in HB.1 to force an r2 versus an r3 fate. Our HCR analyses also indicate a gradual resolution of r2 versus r3 profiles along the AP axis, such that HB.1 cells at the anterior end lose *egr2b* expression first. This may indicate that a morphogen with a source in the anterior hindbrain (such as Fgf produced at the MHB; *Rhinn and Brand, 2001*) acts to bias *vgll3* versus *egr2b* expression within the HB.1 domain, thereby shifting the repressive balance between these TFs and resolving the mixed gene expression profile. The mixed cell identities we observe in the HB.1 domain appear unrelated to the process of 'border sharpening' that takes place during rhombomere formation (*Cooke and Moens, 2002*). In that case, cells of adjacent rhombomeres intermix during boundary formation, which leads to some instances of cells in one rhombomere expressing a gene indicative of the adjacent rhombomere. Such cells are relatively few, usually associated with a rhombomere border, and become sorted into the correct rhombomere based on their expression of *Eph* and *ephrin* cell surface molecules. This contrasts with the mixed identity cells we observe in the HB.1 domain, that are more numerous and not associated with rhombomere boundaries – which have yet to form at this stage. We expect cells in the HB.2 cluster to similarly display a mixed gene expression profile between r5 and r6, but we currently lack robust r6-restricted markers to test this directly. However, we note that 10hpf HB.2 cells contribute to both r5 and r6 in the integrated UMAP (*Figure 6D*) and that the rX cells found in this region of *mafB* mutant embryos give rise to both r5 and r6 (*Moens et al., 1998*; *Moens et al., 1996*) – consistent with HB.2 corresponding to a shared progenitor pool for r5 and r6.

We also detect differential expression of DV markers across all rhombomeres at all stages analyzed (*Figure 1M–P*; *Figure 2E and F*; *Figure 3*, N, O). This finding indicates that early hindbrain cells are simultaneously responding to both DV and AP axial cues – although DV patterning may start prior to AP patterning – further contributing to their mixed molecular identities. Accordingly, DV patterning genes are present in the GRNs for some rhombomeres (e.g. *ntn1* in r6).

At these stages, and at the resolutions tested, we do not detect further subdivision within rhombomeres 1–6 that we deem to be biologically relevant. While this conclusion is consistent with known rhombomere substructures – such as rhombomere border cells – not yet being molecularly distinct at

these stages (*Tambalo et al., 2020*), we cannot exclude the possibility that further analyses – perhaps using supervised clustering – could identify such structures.

# Methods

## Key resources table

| Reagent type (species) or resource | Designation | Source or reference | Identifiers | Additional information |
|---|---|---|---|---|
| Strain, strain background (*Danio rerio*) | AB | Zebrafish International Resource Center | ZL1 | Wild-type zebrafish line |
| Strain, strain background (*Danio rerio*) | TU | Zebrafish International Resource Center | ZL57 | Wild-type zebrafish line |
| Genetic reagent (*Danio rerio*) | TG(hoxb1a:eGFP*um8*) | *Choe et al., 2009* | | Transgenic zebrafish line |
| Genetic reagent (*Danio rerio*) | TG(isl1a:GFP*rw0*) | *Higashijima et al., 2000* | | Transgenic zebrafish line |
| Antibody | Anti-Alcama (mouse monoclonal) | Developmental Studies Hybridoma Bank | ZN-8 | (1:1000) |
| Antibody | Anti-neurofilament antigen (mouse monoclonal) | Developmental Studies Hybridoma Bank | 3A10 | (5 ug/ml) |
| Antibody | Anti-mouse Alexa Fluor 488 (goat polyclonal) | Invitrogen | A11001 | (1:200) |
| Sequence-based reagent | *cyp26c1* HCR probe | Molecular Instruments | PRK068 | 2 pmol |
| Sequence-based reagent | *mafba* HCR probe | Molecular Instruments | PRN449 | 2 pmol |
| Sequence-based reagent | *nab1b* HCR probe | Molecular Instruments | PRQ687 | 2 pmol |
| Sequence-based reagent | *cyp26b1* HCR probe | Molecular Instruments | PRI299 PRR014 | 2 pmol |
| Sequence-based reagent | *egr2b* HCR probe | Molecular Instruments | PRM873 | 2 pmol |
| Sequence-based reagent | *hoxb1a* HCR probe | Molecular Instruments | PRA337 | 2 pmol |
| Sequence-based reagent | *vgll3* HCR probe | Molecular Instruments | PRL721 | 2 pmol |
| Sequence-based reagent | *cdx4* HCR probe | Molecular Instruments | PRN450 | 2 pmol |
| Commercial assay or kit | Chromium Next GEM Single-Cell Multiome ATAC +Gene Expression Reagent Bundle | 10 x Genomics, Inc. | 1000285 | |
| Commercial assay or kit | Chromium Next GEM Chip J Single-Cell Kit | 10 x Genomics, Inc. | 1000234 | |
| Chemical compound, drug | BI protease | Sigma | P5380 | (10 mg/ml) |
| Chemical compound, drug | DEAB | Sigma | D86256 | (5 uM) |
| Chemical compound, drug | SU5402 | Abcam | Ab141368 | 10 uM |
| Software, algorithm | Seurat | CRAN/open source; *Hao et al., 2021* | | https://satijalab.org/seurat/ |
| Software, algorithm | Signac | CRAN/open source; *Stuart et al., 2021* | | https://stuartlab.org/signac/ |

*Continued on next page*

*Continued*

| Reagent type (species) or resource | Designation | Source or reference | Identifiers | Additional information |
|---|---|---|---|---|
| Software, algorithm | DIRECT-NET | Github/open source; *Zhang et al., 2022*; *Zhang, 2023* | | https://github.com/zhanglhbioinfor/DIRECT-NET |
| Software, algorithm | 10 X Genomics Arc 1.0.1 | 10 X Genomics, *Zheng et al., 2017*; *Satpathy et al., 2019* | | https://www.10xgenomics.com/ |
| Software, algorithm | Cytoscape | Cytoscape, *Shannon et al., 2003* | | https://cytoscape.org/ |

## Animals

The Institutional Animal Care and Use Committee (IACUC) of the University of Colorado Medical School approved all procedures involving zebrafish under protocol #870. Wild-type AB (ZL1) and TU (ZL57) zebrafish were obtained from the Zebrafish International Resource Center and reared in our facility. The *TG(hoxb1a:eGFP^um8)* and *TG(isl1a:GFP^rw0)* transgenic lines were reported previously (*Choe et al., 2009*; *Higashijima et al., 2000*). To collect embryos for scMultiome and HCR analyses, one adult male fish and one adult female fish were placed in separate chambers of a 500 mL tank overnight then placed together the following morning. Eggs were collected in 10 cm dishes, immersed in egg water (60 µg/mL Instant Ocean, 0.0002% methylene blue), and maintained in an incubator at 29 °C. Dead and unfertilized eggs were manually removed every 2 hr and were excluded from analyses.

## scMultiome sample preparation

To enrich hindbrain cells, tissue was dissected from the posterior edge of the eye to the level of somite 5 from 25 *TG(hoxb1a:eGFP^um8)* transgenic embryos (that express GFP in rhombomere 4; *Choe et al., 2009*) at 13hpf and 16hpf. Since the hindbrain primordium is difficult to detect at 10hpf, 100 whole embryos were pooled and used at this stage. Dead and unfertilized embryos were excluded from sample preparation. No other animals were excluded, and sample size was selected to achieve sufficient material for analysis and to smooth small variations in developmental stage. Tissue was collected in 1X PBS, dissociated by repeated pipetting through a p1000 tip, and centrifuged at 4°C at 400 G for 5 min followed by resuspension of the pellet in 500 µl of protease solution (10 mg/ml BI protease (Sigma, P5380), 125 U/ml DNase, 2.5 mM EDTA in PBS) for 15 min on ice. Samples were centrifuged at 4°C at 400 G for 5 min, resuspended in 1 ml HBSS + FBS (2%), filtered through a 20 µm cell strainer (pluriSelect, KL-071912), recentrifuged at 4°C at 400 G for 5 min, resuspended in 500 µl of HBSS + FBS (2%) and again filtered through a 20 µm cell strainer. Following centrifugation at 4°C at 400 G for 5 min, the cells were resuspended in 200 µl PBS. For nuclei isolation, we followed the 10 X Genomics recommended protocol with minor modifications. The dissociated cells were centrifuged at 900 G at 4°C for 5 min, resuspended in 100 µl of 0.1 X lysis buffer (1 mM Tris-HCl pH7.4, 1 mM NaCl, 0.3 mM MgCl2, 0.1% BSA, 0.01% Tween-20, 0.01 % NP40, 0.001% Digitonin (Invitrogen, BN2006), 0.1 mM DTT, 0.1 U/µl RNase inhibitor, in nuclease-free water) and incubated on ice for 5 min. Samples were then washed three times by centrifugation at 4°C at 900 G for 10 min and resuspension in 1 ml of chilled wash buffer (10 mM Tris-HCl pH7.4, 10 mM NaCl, 3 mM MgCl2, 1% BSA, 0.1% Tween-20, 1 mM DTT, 1 U/µl RNase inhibitor in nuclease-free water). Finally, nuclei were counted and resuspended in 1 X nuclei buffer (provided in the 10 x Genomics Single-Cell Multiome ATAC Kit A; 1 mM DTT, 1 U/µl RNase inhibitor in nuclease-free water) at a final concentration of approximately 2000–3000 nuclei/µl for sequencing on the 10 X scMultiome platform.

## Single-cell RNA-seq/ATAC-seq analysis

Fastq sequencing files from 10 X Genomics multiomic single-cell RNA-seq and ATAC-seq were processed through Cell Ranger ARC (v1.0.1) with a zebrafish GRCz11 library to obtain UMI gene expression counts and ATAC peak fragment counts. These were analyzed using the standard methods in the Signac (v1.6.0) package in R. Briefly, a Seurat Object was created from the matrix.h5 and fragments.tsv.gz files, annotated with GRCz11.v99.EnsDb, and ATAC peaks corrected by calling with macs2 (v2.2.7.1) and finally filtered to exclude cells with low total RNA expression, low total ATAC counts, high percent mitochondrial gene expression, nucleosome expression >2 or TSS enrichment <1. Gene expression was normalized

with SCTransform and the dimensionality was reduced with PCA. DNA accessibility was processed by performing latent semantic indexing. The Seurat weighted nearest neighbor method was used to compute a neighbor graph which was visualized with UMAP and clusters were annotated based on the expression of marker genes. Per cell motif activities were scored with chromVar. Significantly differentially expressed genes and differential activity motifs were identified for each cluster of cells. The Seurat Objects were integrated by finding the full intersecting ATAC peaks containing peaks in any of the three datasets and creating a new chromatin accessibility assay in each based on the full intersection peak file. The RNA-seq data was integrated with the Seurat V4 integration method and the new chromatin data integrated using Harmony followed by Seurat weighted nearest neighbor method to compute a UMAP and clusters. The cell types and UMAP embeddings for the DEAB-treated 13hpf Seurat Object were predicted using the wild-type 13hpf Seurat Object as a reference following the Seurat V4 reference mapping multimodal reference tutorial. The counts of cells in clusters which expressed a gene were based on cells which had a normalized expression of the gene greater than 0. Dendrograms were calculated with the Seurat BuildClusterTree function. Volcano plots were generated with EnhancedVolcano v1.12.0. A biological replicate of HB13hpf multiomic 10 X Genomics single-cell library was generated and processed identically to the previous single-cell libraries and the two replicates of HB13hpf were integrated and used to calculate GRNs using DIRECT-NET (*Zhang et al., 2022*). DIRECT-NET was written to analyze human single-cell multiomic data for GRNs and relies on matching gene names in the data with JASPAR motif names. In order to utilize the DIRECT-NET package we first converted our scRNAseq gene names to their human orthologs using the human and zebrafish orthology file downloaded from https://zfin.org and modified the scripts slightly to accommodate the use of appropriate motif names. GRNs were visualized with Cytoscape (*Shannon et al., 2003*) using the yfiles radial layout with EGR2-specific networks created by selecting the first neighbors of the EGR2 node and creating a new network from all nodes and edges. scMultiome data has been deposited at GEO under record number GSE223535 and the code used to generate the data for each figure is available at GitHub at https://github.com/rebecaorourke-cu/Sagerstrom_zebrafish_hindbrain, (*O'Rourke, 2023*).

## Hybridization chain reaction analysis

In vivo gene expression was detected by hybridization chain reaction (HCR). HCR probe bundles were purchased from Molecular Instruments (https://www.molecularinstruments.com) under the following lot numbers: *cyp26c1* (PRK068), *mafba* (PRN449), *nab1b* (PRQ687), *cyp26b1* (PRI299 with amplifier B1, PRR014 with amplifier B4); *egr2b* (PRM873); *hoxb1a* (PRA337); *vgll3* (PRL271); *cdx4* (PRN450). Embryos were fixed in 4% paraformaldehyde, dehydrated with 100% methanol, and stored at –20°C until use. Embryos were rehydrated through a series of graded methanol/PBST (PBST: 1XPBS, 0.1% Tween20) washes for 5 min (75% methanol/25% PBST; 50% methanol/50% PBST; 25% methanol/75% PBST; 100% PBST). Embryos were pre-hybridized with a probe hybridization buffer (provided by the manufacturer) for 30 min at 37°C and then placed in a hybridization buffer containing the desired HCR probe sets (2 pmol of each probe set) overnight at 37°C. The next day, the probe solution was removed, and the embryos were washed with probe wash buffer (provided by manufacturer) four times for 15 min at 37°C, followed by washes in 5 X SSCT (5 X SSC, 0.1% Tween20) two times for 5 min. Embryos were pre-treated in an amplification buffer for 30 min at room temperature. Separately, hairpins h1 and h2 were prepared by snap cooling a 3 uM stock, heating at 95°C for 90 s and cooling to room temperature in a dark drawer for 30 min. h1, h2 hairpins were then added to amplification buffer and incubated with embryos overnight in the dark at room temperature. The next day, the hairpin solution was removed, embryos were washed with 5 X SSCT and kept at 4°C until imaging.

HCR analysis in *Figure 1Q–T* to confirm the expression pattern of published genes was carried out on a minimum of 10 wild-type embryos. The number of embryos assayed by HCR for overlap of gene expression in *Figure 6G and H* were as follows: *vgll3/egr2b* – 60 embryos, five biological replicates; *vgll3/nab1b* – 20 embryos, two biological replicates. The number of DEAB-treated embryos assayed by HCR for each probe in *Figure 7H-M* were as follows: *egr2b* – 100 embryos, eight biological replicates; *vgll3* – 40 embryos, four biological replicates; *mafba* – 60 embryos, seven biological replicates; *cyp26c1* – 50 embryos, three biological replicates; *cdx4* – 50 embryos, five biological replicates; *hoxb1a* – 35 embryos, three biological replicates; *cyp26b1* – 45 embryos, six biological replicates. The number of SU5402-treated embryos assayed by HCR for each probe in *Figure 7N–P* were as follows: *egr2b* – 20 embryos, two biological replicates; *hoxb1a* – 30 embryos, three biological replicates; *mafba* – 30

embryos, three biological replicates; *vgll3* – 20 embryos, two biological replicates. An equivalent number of control embryos treated with DMSO (vehicle control) was used. All drug-treated embryos displayed minor variations in the phenotypes shown in *Figure 7H–P* and all vehicle-treated embryos displayed wild-type expression. Sample size was determined based on pilot experiments which demonstrated that all treated embryos showed the reported phenotype. Biological replicate is taken to mean that each replicate was performed on a set of embryos that were treated separately throughout the experiment. The scoring of drug-treated embryos was not blinded.

## Immunohistochemistry

For immunocytochemistry, we used mouse anti-Alcama (1:1000; Developmental Studies Hybridoma Bank) and mouse 3A10 (5 ug/ml; Developmental Studies Hybridoma Bank) primary antibodies. For fluorescence detection, the Alexa Fluro 488 goat anti-mouse (1:200; Invitrogen) secondary antibody was used. Embryos for whole-mount antibody labeling were fixed in 4% AB fix (4% pfa, 8% sucrose, 1 × PBS) overnight at 4 °C and pre-blocked with 10% sheep serum/BSA-1 × PBS for 1 hr at RT. The embryos were incubated in primary antibody for 24 hr at 4 °C, washed semi-continuously with 1 × PBS with 0.2 × Trition (PBSTx) for 2 hr at RT, and then incubated with the secondary antibody for 12 hr at 4 °C, followed by 3 hr of semi-continuous washes with 1 × PBSTx. These embryos were then dissected from the yolk and mounted on bridged cover-slips in 70% glycerol for imaging. Images were imported into Adobe Photoshop and adjustments were made to contrast and level settings. Eight wild-type and eight DEAB-treated embryos were used for each experiment in *Figure 7R and S*. All drug-treated embryos displayed minor variations on the phenotypes shown and all vehicle-treated embryos displayed wild-type neuron formation.

## DEAB and SU5402 treatments

4-Diethylaminobenzaldehyde (DEAB) was used to inhibit retinoic acid signaling. DEAB (Sigma, D86256) was dissolved in DMSO and 5 μM DEAB was used to treat embryos from 4hpf until collected for analysis. SU5402 (Abcam, ab141368) was used to inhibit FGF signaling. SU5402 was dissolved in DMSO and 10 μM used to treat embryos from 4hpf until collected for analysis. Embryos were randomly assigned to treatment versus control. Dead embryos were excluded from analysis.

## Availability of materials and resources

Materials and resources are available by contacting the corresponding author and will be provided free of charge except for reasonable shipping costs.

## Acknowledgements

We thank Douglas Hicks for providing *TG(islet1a:GFP*rw0*)* transgenic embryos.

---

## Additional information

### Funding

| Funder | Grant reference number | Author |
| --- | --- | --- |
| National Institute of Neurological Disorders and Stroke | NS038183 | Charles G Sagerström |

The funders had no role in study design, data collection and interpretation, or the decision to submit the work for publication.

### Author contributions

Yong-Il Kim, Conceptualization, Formal analysis, Validation, Investigation, Visualization, Methodology, Writing – original draft, Writing – review and editing; Rebecca O'Rourke, Data curation, Software, Formal analysis, Visualization, Methodology, Writing – original draft, Writing – review and editing;

Charles G Sagerström, Conceptualization, Formal analysis, Supervision, Funding acquisition, Visualization, Writing – original draft, Project administration, Writing – review and editing

**Author ORCIDs**
Rebecca O'Rourke ⓘ http://orcid.org/0000-0003-1198-6963
Charles G Sagerström ⓘ http://orcid.org/0000-0002-1509-5810

**Ethics**
The Institutional Animal Care and Use Committee (IACUC) of the University of Colorado Medical School approved all procedures involving zebrafish under protocol #870.

**Decision letter and Author response**
Decision letter https://doi.org/10.7554/eLife.87772.sa1
Author response https://doi.org/10.7554/eLife.87772.sa2

## Additional files

**Supplementary files**
• MDAR checklist

**Data availability**
scMultiome data has been deposited at GEO under record number GSE223535. The scMultiome data can also be accessed by users in an interactive format for the three individual timepoints (https://cuanschutz-devbio.shinyapps.io/Sagerstrom_shiny/) or for the integrated data set (https://cuanschutz-devbio.shinyapps.io/Sagerstrom_int3WT_shiny/). The code used to generate the data for each figure is available at GitHub at https://github.com/rebeccaorourke-cu/Sagerstrom_zebrafish_hindbrain., (copy archived at *O'Rourke, 2023*).

The following dataset was generated:

| Author(s) | Year | Dataset title | Dataset URL | Database and Identifier |
|---|---|---|---|---|
| Kim Y-L, O'Rourke R, Sagerstrom CG | 2023 | scMultiome analysis identifies embryonic hindbrain progenitors with mixed rhombomere identities | https://www.ncbi.nlm.nih.gov/geo/query/acc.cgi?acc=GSE223535 | NCBI Gene Expression Omnibus, GSE223535 |

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
