## [Editor Report]

This study transcriptomically profiles the developing zebrafish hindbrain from gastrulation through stages of rhombomere formation in the zebrafish embryo. The transcriptomic data is very thorough and will be a valuable resource to the field.

---

## [Decision Letter]

**Decision letter after peer review:**

Thank you for submitting your article "scMultiome analysis identifies embryonic hindbrain progenitors with mixed rhombomere identities" for consideration by *eLife*. Your article has been reviewed by 3 peer reviewers, and the evaluation has been overseen by a Reviewing Editor and Marianne Bronner as the Senior Editor.

Essential revisions:

1. While the data sets are valuable to the field, the results only provide a small advance in knowledge. In the absence of new functional studies, some of the conclusions are speculative. Thus, the paper should be considered as a Resources and Tools paper rather that an research article.

2. The authors missed an obvious opportunity to examine the periodicity of hindbrain nerves as validation of their interpretations and test whether there is an anatomical manifestation of the altered transcriptome. This would have been easy to replicate with DEAB treated embryos since they already have the transcriptomic data.

3. The authors need to generate a GRN using tool/s designed for multiomic data. Compare and contrast with current GRNs in the literature.

4. The authors should Infer cell trajectories of PHPD cells.

5. A summary figure should be included.

6. Greater discussion or analysis of r1's place in early specification between MHB and HB1, and any potential signs of D-V patterning and sub-rhombomere features at these early stages should be added.

*Reviewer #1 (Recommendations for the authors):*

1. Use the data to generate a GRN for hindbrain segmentation. This could be done using a toolkit such as SCENIC+. The resulting GRN could provide mechanistic insight into early aspects of hindbrain segmentation, which would increase the impact of this work. It would also serve as a very useful point of comparison with hindbrain GRN models described in the literature.

2. Infer cell trajectories to investigate the allocation of PHPD cells into separate rhombomeres. This could provide insight into how PHPDs become subdivided into rhombomeres. A tool such as scVelo or MultiVelo could be used.

3. This manuscript would benefit from a summary figure depicting the key findings and conclusions.

*Reviewer #2 (Recommendations for the authors):*

The goal of this project was to transcriptomically profile the developing zebrafish hindbrain from gastrulation through to rhombomere formation. In that regard, the authors have produced transcriptomic data which will be a valuable resource to the field. However, in its current form, the data is largely descriptive, and in the absence of functional experimentation unfortunately the data tells us very little that's new about the fundamental development or function of the hindbrain and its segmentation into rhombomeres.

In the absence of functional experiments, the authors speculatively assign relevance and significance. In trying to tie their transcriptomic data to dispel the notion and significance of odd versus even numbered rhombomeres the authors really miss the mark and a huge opportunity was lost.

The data the authors present does nothing to counter the developmental value of the concept and importance of odd versus even rhombomeres. In many respects I think the authors don't fully grasp the value or functional anatomical relevance of the odd versus even designation and the wealth of experimental data that revealed how the rhombomeres form as segments (based on differences in cell adhesion, and proliferation at the borders), and the consequences to hindbrain and PNS patterning and development from gene loss-of-function or over expression studies (especially of the Hox, Eph/Eprin and cadherin families), including behavioral changes from converting the character of one even numbered rhombomere to that of another. I'm surprised the authors didn't discuss the work of Chuck Kimmel's lab from 2000 or the principles conveyed in a recent review from Alice Bedois and Robb Krumlauf 2021 that clearly articulate the separation between molecular segmentation and cellular or physical separation and their evolutions. The hindbrain progenitor clusters HB1-3 that the authors identify molecularly, resemble very closely the anatomical or cellular pro-rhombomere A, B, C designations that have been part of the literature since the 1990s. See Osumi-Yamashita et al. 1997 for an example.

The authors' conclusions about egr2 and mafb being primary regulators of r3/r5 and r5/6 identity based on prevalence of binding motifs fits with the known roles of these genes in these segments from loss-of-function studies. Furthermore, the data relating FGF versus RA signaling to regulation of gene expression throughout the hindbrain recapitulates their well know roles. This was however a missed opportunity to functionally test new targets of these signaling pathways to show they have a functional role to play in hindbrain patterning, segmentation, or function with effects on the PNS as well.

The data presented in many figures is too small to be readable and will be lost on the readers. The most obvious examples of this are the dotplots in Figure 1B, 2B, 3C, 7C, S1C, S2C, S3C, S4C

*Reviewer #3 (Recommendations for the authors):*

Thank you for the opportunity to review this work and your patience. My methodological expertise is limited here. However, I find the overall work to be conceptually compelling and worth publishing. I do not see the need for additional experiments, as long as the limitations of their work and approach are discussed as well as how those limitations may be overcome in the future.

The manuscript would be strengthened by consideration of the following points. First, why isn't r1 resolved into a pre-rhombmere cluster? Is it the lack of TF binding motifs unique to r1 or its closer alignment with the MHB? Given engrailed's expression into the MHB, could r1 and midbrain be another proto-cluster? Greater discussion of r1's place in early specification between MHB and HB1 would be helpful.

Dorsal vs ventral patterning can be delineated in this study. Can analysis be done to determine if cells can be further resolved into sub-rhombomere domains such as border vs center or further refinement in the D-V domain be achieved? The neurovascularization of rhombomere follows a notable but consistent pattern across all rhombomeres where sprouting starts in the ventral, A-P center of each rhombomere and follows a circuitous or arched route to connect to the basilar artery (Ulrich et al. 2011; Fujita et al., 2011) indicating a complex but rhombomere defined path for vascularization. Much of this does occur later at 26hpf, but it would be interesting to see if early clusters in 16hpf dataset could be resolved that might ultimately go on to serve as internal landmarks within a rhombomere.

Can border vs center neurons be defined? Beyond boundary or border cells can anterior vs posterior boundary cells be differentiated for a given rhombomere? For example, Fortin et al. 1999 and Coutinho et al. 2004 showed that both the R4 anterior and posterior boundaries were important for the maturation from LF vs HF bursting (likely as part of the chick primordial respiratory rhythm). While the situation regarding hindbrain CPG generation in zebrafish is not fully equivalent to other vertebrates in space and time, the possible delineation of border information here may yield key insights for a number of species. If the data is not of a sufficient density or quality, a discussion on what would be needed to further resolve additional sub-rhombomere features would be welcome.

Line 117 – HCR is not defined at its first use.

---

## [Author Response]

Essential revisions:1. While the data sets are valuable to the field, the results only provide a small advance in knowledge. In the absence of new functional studies, some of the conclusions are speculative. Thus, the paper should be considered as a Resources and Tools paper rather that an research article.

The revised manuscript has been re-submitted as a Tools and Resources paper.

2. The authors missed an obvious opportunity to examine the periodicity of hindbrain nerves as validation of their interpretations and test whether there is an anatomical manifestation of the altered transcriptome. This would have been easy to replicate with DEAB treated embryos since they already have the transcriptomic data.

An analysis of the formation of reticulospinal and branchiomotor neurons in the hindbrain of DEAB-treated embryos is now included in Figure 7Q-S and described in lines 342-348.

3. The authors need to generate a GRN using tool/s designed for multiomic data. Compare and contrast with current GRNs in the literature.

We elected to use the DIRECT-NET tool to derive gene regulatory networks (GRNs) from our multiome data (described in Methods section in lines 589-598). We took advantage of the fact that our Multiome data is the first to successfully distinguishing all hindbrain rhombomeres to generate the first complete set of GRNs for individual rhombomeres (new Figure 5 —figure supplements 1-7). We next demonstrate the utility of these GRNs by highlighting the presence of novel genes in the GRNs relative to prior work and by using them to demonstrate that even closely related rhombomeres (such as r3 and r5) possess clearly distinct GRNs (described in lines 205-223).

4. The authors should Infer cell trajectories of PHPD cells.

The timepoints of our analysis (10hpf, 13hpf and 16hpf) were selected to cover the key steps of rhombomere formation, but the time gaps are too great to permit building continuous trajectories across this time span. Nevertheless, our data can be used to identify transcriptional signatures that change during development (which is a goal of trajectory analysis). We demonstrate this by identifying genes whose expression changes as HB.1 and HB.2 develop in to their derivative rhombomeres (r2 and r3 vs. r5 and r6) in new figures (Figure 6I and Figure 6 —figure supplement 1) and discuss our findings in lines 306-318.

5. A summary figure should be included.

We have included a summary model in Figure 8 and discuss it in lines 470-474.

6. Greater discussion or analysis of r1's place in early specification between MHB and HB1, and any potential signs of D-V patterning and sub-rhombomere features at these early stages should be added.

We do not detect reliable expression of any known r1 markers at 10hpf. Thus, while it is plausible that r1 forms from HB.1 along with r2 and r3 (based on Figure 6D), we cannot say this with certainty, and we have made this clear in lines 452-454. We detect DV patterning at all stages, and this is discussed in lines 129-132, 137-138, 228-229 and 497-502. Using our current parameters, we do not detect evidence for sub-rhombomere features and this is now stated in lines 503-507.

While not requested by the Editors, we have also elected to further clarify how our work relates to the concept of an odd versus even pattern of rhombomere identities. We addressed this point in the discussion of the original submission, but Reviewer 2’s comments suggest that we were not sufficiently clear. There are numerous studies indicating an odd versus even periodicity in terms of cellular/anatomical rhombomere features – cell mixing, neuronal trajectories, neural crest contributions, etc. – but a comprehensive analysis of the molecular basis for such a periodicity has not been possible. Because we were able to identify each rhombomere individually in our scMultiome analysis, we can now carry out direct comparisons of the molecular identities of all rhombomeres. Our finding that there is no wholesale even versus odd molecular pattern is not in disagreement with the presence of cellular/anatomical odd versus even features, instead it raises the question how rhombomeres with unique molecular profiles nevertheless display odd versus even cellular/anatomical features. We have attempted to further clarify this in the text, most notably in the Discussion section in lines 408-431 where we also propose a potential explanation.